# UMIX: Improving Importance Weighting for Subpopulation Shift via Uncertainty-Aware Mixup

**Zongbo Han**[1][*][‡]**, Zhipeng Liang**[2][*][§]**, Fan Yang**[3][*]**, Liu Liu**[3]**, Lanqing Li**[3]**, Yatao Bian**[3]**,**
**Peilin Zhao**[3]**, Bingzhe Wu**[3][†]**, Changqing Zhang**[1][†]**, Jianhua Yao**[3][†]
[1]College of Intelligence and Computing, Tianjin University,
[2] Hong Kong University of Science and Technology, [3]Tencent AI Lab

## Abstract

Subpopulation shift widely exists in many real-world machine learning applications, referring to the training and test distributions containing the same subpopulation groups but varying in subpopulation frequencies. Importance reweighting is a normal way to handle the subpopulation shift issue by imposing constant or adaptive sampling weights on each sample in the training dataset. However, some recent studies have recognized that most of these approaches fail to improve the performance over empirical risk minimization especially when applied to over-parameterized neural networks. In this work, we propose a simple yet practical framework, called uncertainty-aware mixup (UMIX), to mitigate the overfitting issue in over-parameterized models by reweighting the "mixed" samples according to the sample uncertainty. The training-trajectories-based uncertainty estimation is equipped in the proposed UMIX for each sample to flexibly characterize the subpopulation distribution. We also provide insightful theoretical analysis to verify that UMIX achieves better generalization bounds over prior works. Further, we conduct extensive empirical studies across a wide range of tasks to validate the effectiveness of our method both qualitatively and quantitatively. Code is available at this URL.

## 1 Introduction

Empirical risk minimization (ERM) typically faces challenges from distribution shift, which refers to the difference between training and test distributions [3, 27, 61]. One common type of distribution shift is subpopulation shift wherein the training and test distributions consist of the same subpopulation groups but differ in subpopulation frequencies [6, 8]. Many practical research problems (e.g., fairness of machine learning and class imbalance) can all be considered as a special case of subpopulation shift [21, 28, 32]. For example, in the setting of fair machine learning, we train the model on a training dataset with biased demographic subpopulations and test it on an unbiased test dataset [21, 32]. Therefore the essential goal of fair machine learning is to mitigate the subpopulation shift between training and test datasets.

Many approaches have been proposed for solving this problem. Among these approaches, importance weighting (IW) is a classical yet effective technique by imposing static or adaptive weights on each sample when building weighted empirical loss. Therefore each subpopulation group contributes comparably to the final training objective. Specifically, there are normally two ways to achieve importance reweighting. Early works propose to reweight the sample inverse proportionally to the subpopulation frequencies (i.e., static weights) [11, 13, 42, 58, 59, 61], such as class-imbalanced

---

[*]Equal contribution. [‡] Supported by 2021 Tencent Rhino-Bird Research Elite Training Program. [§] Work done during an internship at Tencent AI Lab. [†] Corresponding authors: {`bingzhewu,jianhuayao`}`@tencent.com`, `zhangchangqing@tju.edu.cn`.

36th Conference on Neural Information Processing Systems (NeurIPS 2022).

learning approaches [11, 13, 42]. Alternatively, a more flexible way is to reweight individual samples adaptively according to training dynamics [35, 40, 47, 48, 62, 66, 72, 74]. Distributional robust optimization (DRO) is one of the most representative methods in this line, which minimizes the loss over the worst-case distribution in a neighborhood of the empirical training distribution. A commonly used dual form of DRO can be seen as a special case of importance reweighting wherein the sampling weights are updated based on the current loss [24, 25, 38, 52] in an alternated manner.

However, some recent studies have shown both empirically and theoretically that these IW methods could fail to achieve better worst-case subpopulation performance compared with ERM. Empirically, prior works [10, 58] recognize that various IW methods tend to exacerbate overfitting, which leads to a diminishing effect on stochastic gradient descent (SGD) over training epochs especially when they are applied to over-parameterized neural networks (NNs). Theoretically, previous studies prove that for over-parameterized neural networks, reweighting algorithms do not improve over ERM because their implicit biases are (almost) equivalent [59, 68, 73]. In addition, some prior works also point out that using conventional regularization techniques such as weight decay cannot significantly improve the performance of IW [58].

To this end, we introduce a novel technique called uncertainty-aware mixup (UMIX), by reweighting the mixed samples according to uncertainty within the mini-batch while mitigating overfitting. Specifically, we employ the well-known mixup technique to produce "mixed" augmented samples. Then we train the model on these mixed samples to make sure it can always see "novel" samples thus the effects of IW will not dissipate even at the end of the training epoch. To enforce the model to perform fairly well on all subpopulations, we further efficiently reweight the mixed samples according to uncertainty of the original samples. The weighted mixup loss function is induced by combining the weighted losses of the corresponding two original samples. At a high level, this approach augments training samples in an uncertainty-aware manner, i.e., putting more focus on samples with higher prediction uncertainties that belong to minority subpopulations with high probabilities. We also show UMIX can provide additional theoretical benefit which achieves a tighter generalization bound than weighted ERM [38, 40, 41, 72]. The contributions of this paper are:

- We propose a simple and practical approach called uncertainty-aware mixup (UMIX) to improve previous IW methods by reweighting the mixed samples, which provides a new framework to mitigate overfitting in over-parameterized neural networks.
- Under the proposed framework, we provide theoretical analysis with insight that UMIX can achieve a tighter generalization bound than the weighted ERM.
- We perform extensive experiments on a wide range of tasks, where the proposed UMIX achieves excellent performance in both group-oblivious and group-aware settings.

**Comparison with existing works.** Here, we discuss the key differences between UMIX and other works. In contrast to most IW methods (e.g., CVaR-DRO [38] and JTT [41]), UMIX employs a mixup strategy to improve previous IW methods and mitigate the model overfitting. Among these methods, JTT [41] and LISA [70] are the two most related works to ours. Specifically, JTT provides a two-stage optimization framework in which an additional network is used for building the error set, and then JTT upweights samples in the error set in the following training stage. Besides, LISA also modifies mixup for improving model robustness against distribution shift. However, LISA intuitively mixes the samples within the same subpopulation or same label thus it needs additional subpopulation information. In contrast to them, UMIX introduces sample weights into the vanilla mixup strategy by quantitatively measuring the sample uncertainties without subpopulation information. In addition, our work is orthogonal to LISA, i.e., we can use our weight building strategy to improve LISA's performance. In practice, our method consistently outperforms previous approaches that do not use subpopulation information and even achieves quite competitive performance to those methods which leverage subpopulation information. We also provide theoretical analysis to explain why UMIX works better than the weighted ERM [38, 40, 41, 72].

## 2 Related Work

### 2.1 Importance weighting

To improve the model robustness against subpopulation shift, importance weighting (IW) is a classical yet effective technique by imposing static or adaptive weight on each sample and then building

weighted empirical loss. Therefore each subpopulation group can have a comparable strength in the final training objective. Specifically, there are typically two ways to achieve importance reweighting, i.e., using static or adaptive importance weights.

**Static methods**. The naive reweighting approaches perform static reweighting based on the distribution of training samples [11, 13, 42, 58, 59, 61]. Their core motivation is to make different subpopulations have a comparable contribution to the training objective by reweighting. Specifically, the most intuitive way is to set the weight of each sample to be inversely proportional to the number of samples in each subpopulation [58, 59, 61]. Besides, there are some methods to obtain sample weights based on the effective number of samples [13], subpopulation margins [11], and Bayesian networks [42].

**Adaptive methods**. In contrast to the above static methods, a more essential way is to assign each individual sample an adaptive weight that can vary according to training dynamics [35, 40, 47, 48, 62, 66, 72, 74]. Distributional robust optimization (DRO) is one of the most representative methods in this line, which minimizes the loss over the worst-case distribution in a neighborhood of the empirical training distribution. A commonly-used dual form of DRO can be considered as a special case of importance reweighting wherein the sampling weights are updated based on the current loss [24, 25, 38, 52] in an alternated manner. For example, in the group-aware setting (i.e., we know each sample belongs to which subpopulation), GroupDRO [58] introduces an online optimization algorithm to update the weights of each group. In the group-oblivious setting, [35, 47, 48, 66] model the problem as a (regularized) minimax game, where one player aims to minimize the loss by optimizing the model parameters and another player aims to maximize the loss by assigning weights to each sample.

## 2.2 Uncertainty quantification

The core of our method is based on the high-quality uncertainty quantification of each sample. There are many approaches proposed for this goal. The uncertainty of deep learning models includes epistemic (model) uncertainty and aleatoric (data) uncertainty [30]. To obtain the epistemic uncertainty, Bayesian neural networks (BNNs) [15, 30, 45, 53] have been proposed which replace the deterministic weight parameters of model with distribution. Unlike BNNs, ensemble-based methods obtain the epistemic uncertainty by training multiple models and ensembling them [2, 22, 26, 36]. Aleatoric uncertainty focuses on the inherent noise in the data, which usually is learned as a function of the data [30, 37, 54]. Uncertainty quantification has been successfully equipped in many fields such as multimodal learning [19, 20, 44], multitask learning [14, 31], and reinforcement learning [29, 39]. Unlike previous methods, our method focuses on estimating the epistemic uncertainty of training samples with subpopulation shift and upweighting uncertain samples, thereby improving the performance of minority subpopulations with high uncertainty.

## 3   Method

In this section, we introduce technical details of UMIX. The key idea of UMIX is to exploit uncertainty information to upweight mixed samples, and thus can encourage the model to perform uniformly well on all subpopulations. We first introduce the basic procedure of UMIX and then present how to provide high-quality uncertainty estimations which is the fundamental block of UMIX.

### 3.1   Background

The necessary background and notations are provided here. Let the input and label space be $\mathcal{X}$ and $\mathcal{Y}$ respectively. Given training dataset $\mathcal{D}$ with $N$ training samples $\{(x_i, y_i)\}_{i=1}^{N}$ i.i.d. sampled from a probability distribution $P$. We consider the setting that the training distribution $P$ is a mixture of $G$ predefined subpopulations, i.e., $P = \sum_{g=1}^{G} k_g P_g$, where $k_g$ and $P_g$ denote the $g$-th subpopulation's proportion and distribution respectively. Our goal is to obtain a model $f_\theta : \mathcal{X} \to \mathcal{Y}$ parameterized by $\theta \in \Theta$ that performs well on all subpopulations.

The well-known empirical risk minimization (ERM) algorithm doesn't consider the subpopulations and minimizes the expected risk $\mathbb{E}[\ell(\theta, x_i, y_i)]$, where $\ell$ denotes the loss function. This leads to the model paying more attention to the majority subpopulations in the training set and resulting in poor performance on the minority subpopulations. For example, the ERM-based models may learn

spurious correlations that exist in majority subpopulations but not in minority subpopulations [58]. The proposed method aims to learn a model that is robust against subpopulation shift by importance weighting.

Previous works on improving subpopulation shift robustness investigate several different settings, i.e., group-aware and group-oblivious [41, 58, 72]. Most of the previous works have assumed that the group label is available during training [58, 70]. This is called the group-aware setting. However, due to some reasons, we may not have training group labels. For example, in many real applications, it's hard to extract group label information. Meanwhile, the group label information may not be available due to privacy concerns. This paper studies the group-oblivious setting, which cannot obtain group information for each example at training time. This requires the model to identify underperforming samples and then pay more attention to them during training.

### 3.2 Importance-weighted mixup

UMIX employs an aggressive data augmentation strategy called uncertainty-aware mixup to mitigate overfitting. Specifically, vanilla mixup [75, 76] constructs virtual training examples (i.e., mixed samples) by performing linear interpolations between data/features and corresponding labels as:

$$\widetilde{x}_{i,j} = \lambda x_i + (1 - \lambda)x_j, \ \widetilde{y}_{i,j} = \lambda y_i + (1 - \lambda)y_j, \tag{1}$$

where $(x_i, y_i), (x_j, y_j)$ are two samples drawn at random from empirical training distribution and $\lambda \in [0, 1]$ is usually sampled from a beta distribution. Then vanilla mixup optimizes the following loss function:

$$\mathbb{E}_{\{(x_i,y_i),(x_j,y_j)\}}[\ell(\theta, \widetilde{x}_{i,j}, \widetilde{y}_{i,j})]. \tag{2}$$

When the cross entropy loss is employed, Eq. 2 can be rewritten as:

$$\mathbb{E}_{\{(x_i,y_i),(x_j,y_j)\}}[\lambda \ell(\theta, \widetilde{x}_{i,j}, y_i) + (1 - \lambda)\ell(\theta, \widetilde{x}_{i,j}, y_j)]. \tag{3}$$

Eq. 3 can be seen as a linear combination (mixup) of $\ell(\theta, \widetilde{x}_{i,j}, y_i)$ and $\ell(\theta, \widetilde{x}_{i,j}, y_j)$. Unfortunately, since vanilla mixup doesn't consider the subpopulations with poor performance, it has been shown experimentally to be non-robust against subpopulation shift [70]. To this end, we introduce a simple yet effective method called UMIX, which further employs a weighted linear combination of the original loss based on Eq. 3 to encourage the learned model to pay more attention to samples with poor performance.

In contrast to previous IW methods, the importance weights of UMIX are used on the mixed samples. To do this, we first estimate the uncertainty of each sample and then use this quantity to construct the importance weight (i.e., the higher the uncertainty, the higher the weight, and vice versa). For the $i$-th sample $x_i$, we denote its importance weight as $w_i$. Once we obtain the importance weight, we can perform weighted linear combination of $\ell(\theta, \widetilde{x}_{i,j}, y_i)$ and $\ell(\theta, \widetilde{x}_{i,j}, y_j)$ by:

$$\mathbb{E}_{\{(x_i,y_i),(x_j,y_j)\}}[w_i\lambda \ell(\theta, \widetilde{x}_{i,j}, y_i) + w_j(1 - \lambda)\ell(\theta, \widetilde{x}_{i,j}, y_j)], \tag{4}$$

where $w_i$ and $w_j$ denote the importance weight of the $i$-th and $j$-th samples respectively. In practice, to balance the UMIX and normal training, we set a hyperparameter $\sigma$ that denotes the probability to apply UMIX. The whole training pseudocode for UMIX is shown in Algorithm 1.

---

**Algorithm 1:** The training pseudocode of UMIX.

**Input:** Training dataset $\mathcal{D}$ and the corresponding importance weights $\mathbf{w} = [w_1, \cdots, w_N]$, hyperparameter $\sigma$ to control the probability of doing UMIX, and parameter $\alpha$ of the beta distribution;

1 **for** *each iteration* **do**
2      Obtain training samples $(x_i, y_i), (x_j, y_j)$ and the corresponding weight $w_i, w_j$;
3      Sample $p \sim \text{Uniform}(0,1)$;
4      **if** $p < \sigma$ **then** Sample $\lambda \sim Beta(\alpha, \alpha)$; **else** $\lambda = 0$;
5      Obtain the mixed input $\widetilde{x}_{i,j}$ where $\widetilde{x}_{i,j} = \lambda x_i + (1 - \lambda)x_j$;
6      Obtain the loss of the model with $w_i\lambda \ell(\theta, \widetilde{x}_{i,j}, y_i) + w_j(1 - \lambda)\ell(\theta, \widetilde{x}_{i,j}, y_j)$;
7      Update model parameters $\theta$ to minimize loss with an optimization algorithm.

---

### 3.3 Uncertainty-aware importance weights

Now we present how to obtain the uncertainty-aware training importance weights. In the group-oblivious setting, the key to obtaining importance weights is to find samples with high uncertainty. For example, DRO-based algorithms construct the uncertainty set with the current loss [24, 25, 38, 52]. It has been shown experimentally that the uncertain samples found in this way are constantly changing during training [41], resulting in these methods not always upweighting the minority subpopulations. Therefore, we introduce a sampling-based stable uncertainty estimation to better characterize the subpopulation shift.

Given a well trained neural classifier $f_\theta : \mathcal{X} \to \mathcal{Y}$ that could produce the predicted class $\hat{f}_\theta(x)$, a simple way to obtain the uncertainty of a sample is whether the sample is correctly classified. However, as pointed out in previous work [36], a single model cannot accurately characterize the sampling uncertainty. Therefore, we propose to obtain the uncertainty through Bayesian sampling from the model posterior distribution $p(\theta; \mathcal{D})$. Specifically, given a sample $(x_i, y_i)$, we define the training uncertainty as:

$$u_i = \int \kappa(y_i, \hat{f}_\theta(x_i)) p(\theta; \mathcal{D}) d\theta, \text{where } \kappa(y_i, \hat{f}_\theta(x_i)) = \begin{cases} 0, & \text{if } y_i = \hat{f}_\theta(x_i) \\ 1, & \text{if } y_i \neq \hat{f}_\theta(x_i) \end{cases}. \tag{5}$$

Then, we can obtain an approximation of Eq. 5 with $T$ Monte Carlo samples as $u_i \approx \frac{1}{T} \sum_{t=1}^{T} \kappa(y_i, \hat{f}_{\theta_t}(x_i))$, where $\theta_t \in \Theta$ can be obtained by minimizing the expected risk.

In practice, sampling $\{\theta_t\}_{t=1}^{T}$ from the posterior (i.e., $\theta_t \sim p(\theta; \mathcal{D})$) is computationally expensive and sometimes even intractable since multiple training models need to be built or extra approximation errors need to be introduced. Inspired by a recent Bayesian learning paradigm named SWAG [46], we propose to employ the information from the historical training trajectory to approximate the sampling process. More specifically, we train a model with ERM and save the prediction results $\hat{f}_{\theta_t}(x_i)$ of each sample on each iteration epoch $t$. Then, to avoid the influence of inaccurate predictions at the beginning of training, we estimate uncertainty with predictions after training $T_s - 1$ epochs with:

$$u_i \approx \frac{1}{T} \sum_{t=T_s}^{T_s+T} \kappa(y_i, \hat{f}_{\theta_t}(x_i)). \tag{6}$$

We empirically show that the proposed approximation could obtain reliable uncertainty in Sec. B.4 of the Appendix.

To obtain reasonable importance weights, we assume that the samples with high uncertainty should be given a higher weight and vice versa. Therefore a reasonable importance weight could be linearly positively related to the corresponding uncertainty,

$$w_i = \eta u_i + c, \tag{7}$$

where $\eta \in \mathbb{R}_+$ is a hyperparameter and $c \in \mathbb{R}_+$ is a constant that keeps the weight to be positive. In practice, we set $c$ to 1. The whole process for obtaining training importance weights is shown in Algorithm 2.

---

**Algorithm 2:** The process for obtaining training importance weights.

---

**Input:** Training dataset $\mathcal{D}$, sampling start epoch $T_s$, the number of sampling $T$, and upweight hyperparameter $\eta$ ;
**Output:** The training importance weights $\mathbf{w} = [w_1, \cdots, w_n]$;

1 **for** *each iteration* **do**
2      Train $f_\theta$ by minimizing the expected risk $\mathbb{E}\{\ell(\theta, x_i, y_i)\}$;
3      Save the prediction results $\{\hat{f}_{\theta_t}(x_i)\}_{i=1}^{N}$ of the current epoch $t$;
4 Obtain the uncertainty of each sample with $u_i \approx \frac{1}{T} \sum_{t=T_s}^{T_s+T} \kappa(y_i, \hat{f}_{\theta_t}(x_i))$;
5 Obtain the importance weight of each sample with $w_i = \eta u_i + c$.

---

**Remark.** Total uncertainty can be divided into epistemic and aleatoric uncertainty [30]. In the proposed method, the samples are weighted only based on epistemic uncertainty by sampling from

the model on the training trajectory, which can be seen as sampling from the model posterior in a more efficient way. What's more, we consider that the training samples do not contain the inherent noise (aleatoric uncertainty) since it is usually intractable to distinguish between noisy samples and minority samples from data with subpopulation shifts.

**Rethink why this estimation approach could work?** Recent work has empirically shown that compared with the hard-to-classify samples, the easy-to-classify samples are learned earlier during training [18]. Meanwhile, the hard-to-classify samples are also more likely to be forgotten by the neural networks [64]. The frequency with which samples are correctly classified during training can be used as supervision information in confidence calibration [51]. Snapshot performs ensemble learning on several local minima models along the optimization path [26]. The proposed method is also inspired by these observations and algorithms. During training, samples from the minority subpopulations are classified correctly less frequently, which corresponds to higher training uncertainty. On the other hand, samples from the majority subpopulations will have lower training uncertainty due to being classified correctly more often. In Sec. B.5 of the Appendix, we show the accuracy of different subpopulations during training to empirically validate our claim. Meanwhile, we explain in detail why the uncertainty estimation based on historical information is chosen in Sec. C of the Appendix.

# 4 Experiments

In this section, we conduct experiments on multiple datasets with subpopulation shift to answer the following questions. Q1 Effectiveness (I). In the group-oblivious setting, does the proposed method outperform other algorithms? Q2 Effectiveness (II). How does UMIX perform without the group labels in the validation set? Q3 Effectiveness (III). Although our method does not use training group labels, does it perform better than the algorithms using training group labels? Q4 Effectiveness (IV). Can UMIX improve the model robustness against domain shift where the training and test distributions have different subpopulations. Q5 Qualitative analysis. Are the obtained uncertainties of the training samples trustworthy? Q6 Ablation study. What is the key factor of performance improvement in our method?

## 4.1 Setup

We briefly present the experimental setup here, including the experimental datasets, evaluation metrics, model selection, and comparison methods. Please refer to Sec. B in Appendix for more detailed setup.

**Datasets**. We perform experiments on three datasets with multiple subpopulations, including Waterbirds [58], CelebA [43] and CivilComments [9]. We also validate UMIX on domain shift scenario which is a more challenging distribution shift problem since there are different subpopulations between training and test data. Hence, we conduct experiments on a medical dataset called Camelyon17 [5, 33] that consists of pathological images from five different hospitals. The training data is drawn from three hospitals, while the validation and test data are sampled from other hospitals.

**Evaluation metrics**. To be consistent with existing works [33, 56, 70], we report the average accuracy of Camelyon17 over 10 different random seeds. On other datasets, we repeat experiments over 3 times and report the average and worst-case accuracy among all subpopulations. The trade-off between the average and worst-case accuracy is a well-known challenge [21]. In this paper, we lay emphasis on worst-case accuracy, which is more important than average accuracy in some application scenarios. For example, in fairness-related applications, we should pay more attention to the performance of the minority groups to reduce the gap between the majority groups and ensure the fairness of the machine learning decision system.

**Model selection**. Following prior works [41, 72], we assume the group labels of validation samples are available and select the best model based on worst-case accuracy among all subpopulations on the validation set. We also conduct model selection based on the average accuracy to show the impact of validation group label information in our method.

**Comparisons in the group-oblivious setting**. Here we list the baselines used in the group-oblivious setting. (1) ERM trains the model using standard empirical risk minimization. (2) Focal loss [40] downweights the well-classified examples' loss according to the current classification confidences.

(3) DRO-based methods including CVaR-DRO, $\chi^2$-DRO [38], CVaR-DORO and $\chi^2$-DORO [72] minimize the loss over the worst-case distribution in a neighborhood of the empirical training distribution. (4) JTT [41] constructs an error set and upweights the samples in the error set to improve the worst-case performance among all subpopulations.

**Comparison in the group-aware setting**. To better demonstrate the performance of the proposed method, we compare our method with multiple methods that use training group labels, including IRM [3], IB-IRM [1], V-REx [34], CORAL [63], Group DRO [58], DomainMix [69], Fish [60], and LISA [70].

**Mixup-based comparison methods**. We compare our method with vanilla mixup and in-group mixup, where vanilla mixup is performed on any pair of samples and in-group mixup is performed on the samples with the same labels and from the same subpopulations.

## 4.2  Experimental results

We present experimental results and discussions to answer the above-posed questions.

**Q1 Effectiveness** (I). Since our algorithm does not need training group labels, thus we conduct experiments to verify its superiority over current group-oblivious algorithms. The experimental results are shown in Table 1 and we have the following observations: (1) The proposed UMix achieves the best worst-case accuracy on all three datasets. For example, for the CelebA dataset, UMix achieves worst-case accuracy of 85.3%, while the second-best is 81.1%. (2) ERM consistently outperforms other methods in terms of average accuracy. However, it typically comes with the lowest worst-case accuracy. The underlying reason is that the dominance of the majority subpopulations during training leads to poor performance of the minority subpopulations. (3) UMix shows competitive average accuracy compared to other methods. For example, on CelebA, UMix achieves the average accuracy of 90.1%, which outperforms all other IW/DRO methods.

**Q2 Effectiveness** (II). We conduct the evaluation on the Waterbirds and CelebA datasets without using the validation set group label information. Specifically, after each training epoch, we evaluate the performance of the current model on the validation set and save the model with the best average accuracy. Finally, we test the performance of the saved model on the test set. The experimental results are shown in Table 2. From the experimental results, we can observe that when the validation set group information is not used, the worst-case accuracy of our method drops a little while the average accuracy improves a little.

**Q3 Effectiveness** (III). We further conduct comparisons with algorithms that require training group labels. The comparison results are shown in Table 3. According to the experimental results, it is observed that the performance from our UMix without using group label is quite competitive compared with these group-aware algorithms. Specifically, benefiting from the uncertainty-aware mixup, UMix usually performs in the top three in terms of both average and worst-case accuracy. For example, on WaterBirds, UMix achieves the best average accuracy of 93.0% and the second-best worst-case accuracy of 90.0%.

Table 1: Comparison results with other methods in the group-oblivious setting. The best results are in bold and blue. Full results with standard deviation are in the Table 6 in Appendix.

|  | Waterbirds | | CelebA | | CivilComments | | Camelyon17 |
|---|---|---|---|---|---|---|---|
|  | Avg. | Worst | Avg. | Worst | Avg. | Worst | Avg. |
| ERM | **97.0%** | 63.7% | **94.9%** | 47.8% | **92.2%** | 56.0% | 70.3% |
| Focal Loss [40] | 87.0% | 73.1% | 88.4% | 72.1% | 91.2% | 60.1% | 68.1% |
| CVaR-DRO [38] | 90.3% | 77.2% | 86.8% | 76.9% | 89.1% | 62.3% | 70.5% |
| CVaR-DORO [72] | 91.5% | 77.0% | 89.6% | 75.6% | 90.0% | 64.1% | 67.3% |
| $\chi^2$-DRO [38] | 88.8% | 74.0% | 87.7% | 78.4% | 89.4% | 64.2% | 68.0% |
| $\chi^2$-DORO [72] | 89.5% | 76.0% | 87.0% | 75.6% | 90.1% | 63.8% | 68.0% |
| JTT [41] | 93.6% | 86.0% | 88.0% | 81.1% | 90.7% | 67.4% | 69.1% |
| Ours | 93.0% | **90.0%** | 90.1% | **85.3%** | 90.6% | **70.1**% | **75.1%** |

Table 2: Experimental results when the group labels in the validation set are available or not.

| Group labels in validation set? | Waterbirds | | CelebA | |
|---|---|---|---|---|
| | Average ACC | Worst-case ACC | Average ACC | Worst-case ACC |
| Yes | 93.00% | 90.00% | 90.10% | 85.30% |
| No | 93.60% | 88.90% | 90.40% | 84.60% |

Table 3: Comparison results with the algorithms **using training group labels** (Our method is not dependent on this type of information). Results of baseline models are from [70]. The best three results are in bold brown or bold blue and the color indicates whether the training group labels are used. Full results with standard deviation are in the Table 7 in Appendix.

| | Group labels in train set? | Waterbirds | | CelebA | | CivilComments | | Cam17 |
|---|---|---|---|---|---|---|---|---|
| | | Avg. | Worst | Avg. | Worst | Avg. | Worst | Avg. |
| IRM [3] | Yes | 87.5% | 75.6% | **94.0%** | 77.8% | 88.8% | 66.3% | 64.2% |
| IB-IRM [1] | Yes | 88.5% | 76.5% | **93.6%** | 85.0% | 89.1% | 65.3% | 68.9% |
| V-REx [34] | Yes | 88.0% | 73.6% | 92.2% | **86.7%** | **90.2%** | 64.9% | 71.5% |
| CORAL [63] | Yes | 90.3% | 79.8% | **93.8%** | 76.9% | 88.7% | 65.6% | 59.5% |
| GroupDRO [58] | Yes | **91.8%** | **90.6%** | 92.1% | **87.2%** | 89.9% | 70.0% | 68.4% |
| DomainMix [69] | Yes | 76.4% | 53.0% | 93.4% | 65.6% | **90.9%** | 63.6% | 69.7% |
| Fish [60] | Yes | 85.6% | 64.0% | 93.1% | 61.2% | 89.8% | **71.1%** | **74.7%** |
| LISA [70] | Yes | **91.8%** | **89.2%** | 92.4% | **89.3%** | 89.2% | **72.6%** | **77.1%** |
| Ours | **No** | **93.0%** | **90.0%** | 90.1% | 85.3% | **90.6%** | **70.1%** | **75.1%** |

**Q4 Effectiveness** (IV). We conduct comparison experiments on Camelyon17 to investigate the effectiveness of our algorithm under the domain shift scenario. The experimental results are shown in the last column of Table 1 and Table 3 respectively. In the group-oblivious setting, the proposed method achieves the best average accuracy on Camelyon17 as shown in Table 1. For example, UMɪx achieves the best average accuracy of 75.1% while the second is 70.3%. Meanwhile, in Table 3, benefiting from upweighting the mixed samples with poor performance, our method achieves a quite competitive generalization ability on Camelyon17 compared with other algorithms using training group labels.

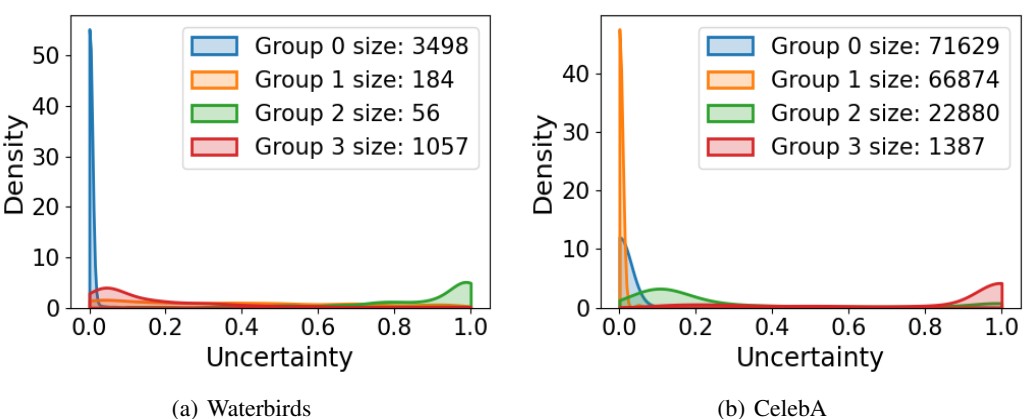

(a) Waterbirds                                    (b) CelebA

Figure 1: Visualization of the obtained uncertainty with kernel density estimation on Waterbirds and CelebA datasets, where group size refers to the sample number of the group.

**Q5 Qualitative analysis**. To intuitively investigate the rationality of the estimated uncertainty, we visualize the density of the uncertainty for different groups with kernel density estimation. As shown in Fig. 1, the statistics of estimated uncertainty is basically correlated to the training sample size of each group. For example, on Waterbirds and CelebA, the average uncertainties of minority groups are much higher, while those of majority groups are much lower.

**Q6 Ablation study**. Finally, we conduct the ablation study in comparison with vanilla mixup and in-group mixup. The experimental results are shown in Table 4. Compared with ERM, vanilla mixup cannot significantly improve worst-case accuracy. After using the group label, the in-group mixup slightly improves the worst-case accuracy compared to ERM. The possible reason is that mixup-based methods do not increase the influence of minority subpopulations in the model objective function. Although our method does not use the group label of the training samples, our method can still significantly improve the worst-case accuracy.

Table 4: Comparison with ERM and mixup based methods. Results of baseline models are from [70]. The best results are in bold brown or bold blue and the color indicates whether the training group labels are used. Full results with standard deviation are in the Table 8 in Appendix.

| | Group labels in train set? | Waterbirds Avg. | Worst | CelebA Avg. | Worst | CivilComments Avg. | Worst | Cam17 Avg. |
|---|---|---|---|---|---|---|---|---|
| ERM | No | **97.0%** | 63.7% | 94.9% | 47.8% | **92.2%** | 56.0% | 70.3% |
| vanilla mixup | No | 81.0% | 56.2% | **95.8%** | 46.4% | 90.8% | 67.2% | 71.2% |
| in-group mixup | Yes | 88.7% | 68.0% | 95.2% | 58.3% | 90.8% | 69.2% | **75.5%** |
| Ours | No | 93.0% | **90.0%** | 90.1% | **85.3%** | 90.6% | **70.1%** | 75.1% |

## 5 Theory

In this section, we provide a theoretical understanding of the generalization ability for UMIX. At a high level, we prove that our method can achieve a better generalization error bound than traditional IW methods without using mixup. For simplicity, our analysis focuses on generalized linear model (GLM). The roadmap of our analysis is to first approximate the mixup loss and then study the generalization bound from a Rademacher complexity perspective. To introduce the theoretical framework, we first present the basic settings.

**Basic settings.** Our analysis mainly focuses on GLM model classes whose loss function $\ell$ follows $\ell(\theta, x, y) = A(\theta^\top x) - y\theta^\top x$, where $x \in \mathbb{R}^d$ is the input, $\theta \in \mathbb{R}^d$ is the parameter, $y \in \mathbb{R}$ is the label and $A(\cdot)$ is the log-partition function.

Recall the setting of subpopulation shift, we assume that the population distribution $P$ consists of $G$ different subpopulations with the $g$-th subpopulation's proportion being $k_g$ and the $g$-th subpopulation follows the distribution $P_g$. Specifically, we have $P = \sum_{g=1}^G k_g P_g$. Then we denote the covariance matrix for the $g$-th subpopulation as $\Sigma_X^g = \mathbb{E}_{(x,y)\sim P_g}[xx^\top]$. For simplicity, we consider the case where a shared weight $w_g$ is assigned to all samples from the $g$-th subpopulation. The main goal of our theoretical analysis is to characterize the generalization ability of the model learned using Algorithm 1. Formally, we focus on analyzing the upper bound of the weighted generalization error defined as:

$$\mathrm{GError}(\theta) = \mathbb{E}_{(x,y)\sim P}[w(x,y)\ell(\theta,x,y)] - \frac{1}{N}\sum_{i=1}^N w(x_i,y_i)\ell(\theta,x_i,y_i),$$

where the function $w(x,y)$ is the weighted function to return the weight of the subpopulation to which the sample $(x, y)$ belongs.

First of all, we present our main result in this section. The main theorem of our analysis provides a subpopulation-heterogeneity dependent bound for the above generalization error. This theorem is formally presented as:

**Theorem 5.1.** *Suppose $A(\cdot)$ is $L_A$-Lipschitz continuous, then there exists constants $L, B > 0$ such that for any $\theta$ satisfying $\theta^\top \Sigma_X \theta \leq \gamma$, the following holds with a probability of at least $1 - \delta$,*

$$\mathrm{GError}(\theta) \leq 2L \cdot L_A \cdot (\max\{(\frac{\gamma(\delta/2)}{\rho})^{1/4}, (\frac{\gamma(\delta/2)}{\rho})^{1/2}\} \cdot \sqrt{\frac{\mathrm{rank}(\Sigma_X)}{n}}) + B\sqrt{\frac{\log(2/\delta)}{2n}},$$

*where $\gamma(\delta)$ is a constant dependent on $\delta$, $\Sigma_X = \sum_{g=1}^G k_g w_g \Sigma_X^g$ and $\rho$ is some constant related to the data distribution, which will be formally introduced in Assumption 5.1.*

We will show later that the output of our Algorithm 1 can satisfy the constraint $\theta^\top \Sigma_X \theta \leq \gamma$ and thus Theorem 5.1 can provide a theoretical understanding of our algorithm. In contrast to weighted ERM, the bound improvement of UMIX is on the red term which can partially reflect the heterogeneity of the training subpopulations. Specifically, the red term would become $\sqrt{d/n}$ in the weighted ERM setting (see more detailed theoretical comparisons in Appendix). Thus our bound can be tighter when the intrinsic dimension of data is small (i.e., rank$(\Sigma_X) \ll d$).

The proof of Theorem 5.1 follows this roadmap: (1) We first show that the model learned with UMIX can fall into a specific hypothesis set $\mathcal{W}_\gamma$. (2) We analyze the Rademacher complexity of the hypothesis set and obtain its complexity upper bound (Lemma A.3). (3) Finally, we can characterize the generalization bound by using complexity-based learning theory [7] (Theorem 8). More details of the proof can be found in Appendix.

As we discuss in Appendix, the weighted mixup can be seen as an approximation of a regularization term $\frac{C}{n}[\sum_{i=1}^n w_i A''(x_i^\top \theta)]\theta^\top \widehat{\Sigma}_X \theta$ for some constant $C$ compared with the non-mixup algorithm, which motivates us to study the following hypothesis space

$$\mathcal{W}_\gamma := \{x \to \theta^\top x, \text{such that } \theta \text{ satisfying } \mathbb{E}_{x,y}[w(x,y)A''(x^\top \theta)]\theta^\top \Sigma_X \theta \leq \gamma\},$$

for some constant $\gamma$.

To further derive the generalization bound, we also need the following assumption, which is satisfied by general GLMs when $\theta$ has bounded $\ell_2$ norm and it is adopted in, e.g., [4, 76].

**Assumption 5.1** ($\rho$-retentive). *We say the distribution of $x$ is $\rho$-retentive for some $\rho \in (0, 1/2]$ if for any non-zero vector $v \in \mathbb{R}^d$ and given the event that $\theta \in \mathcal{W}_\gamma$ where the $\theta$ is output by our Algorithm 1, we have*

$$\mathbb{E}_x^2[A''(x^\top v)] \geq \rho \cdot \min\{1, \mathbb{E}_x(v^\top x)^2\}.$$

Finally, we can derive the Rademacher complexity of the $\mathcal{W}_\gamma$ and the proof of Theorem 5.1 is obtained by combining Lemma A.3 and the Theorem 8 of [7].

**Lemma 5.1.** *Assume that the distribution of $x_i$ is $\rho$-retentive, i.e., satisfies the assumption 5.1. Then the empirical Rademacher complexity of $\mathcal{W}_r$ satisfies*

$$Rad(\mathcal{W}_r, \mathcal{S}) \leq \max\{(\frac{\gamma(\delta)}{\rho})^{1/4}, (\frac{\gamma(\delta)}{\rho})^{1/2}\} \cdot \sqrt{\frac{rank(\Sigma_X)}{n}},$$

*with probability at least $1 - \delta$.*

## 6   Conclusion

In this paper, we propose a novel method called UMIX to improve the model robustness against subpopulation shift. We propose a simple yet reliable approach to estimate the sample uncertainties and integrate them into the mixup strategy so that UMIX can mitigate the overfitting thus improving over prior IW methods. Our method consistently outperforms previous approaches on commonly-used benchmarks. Furthermore, UMIX also shows the theoretical advantage that the learned model comes with subpopulation-heterogeneity dependent generalization bound. In the future, how to leverage subpopulation information to improve UMIX can be a promising research direction.

## Acknowledgements

This work was supported in part by the National Key Research and Development Program of China under Grant 2019YFB2101900, the National Natural Science Foundation of China (61976151, 61925602, 61732011).

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
