# Appendix

## Contents

## A Proofs

In this appendix, we prove the Theorem 5.1 in Section 5. We consider the following optimization objective, which is the expected version of our weighted mixup loss (Equation 4).

$$L_n^{\mathrm{mix}}(\theta, S) = \frac{1}{n^2} \sum_{i,j=1}^{n} \mathbb{E}_{\lambda \sim D_\lambda}[\lambda w_i l(\theta, \tilde{x}_{i,j}, y_i) + (1-\lambda) w_j l(\theta, \tilde{x}_{i,j}, y_j)],$$

where the loss function we consider is $l(\theta, x, y) = h(f_\theta(x)) - y f_\theta(x)$ and $h(\cdot)$ and $f_\theta(\cdot)$ for all $\theta \in \Theta$ are twice differentiable. We compare it with the standard weighted loss function

$$L_n^{std}(\theta, S) = \frac{1}{n} \sum_{i=1}^{n} w_i[h(f_\theta(x_i)) - y_i f_\theta(x_i)].$$

**Lemma A.1.** *The weighted mixup loss can be rewritten as*

$$L_n^{mix}(\theta, S) = L_n^{std}(\theta, S) + \sum_{i=1}^{3} \mathcal{R}_i(\theta, S) + \mathbb{E}_{\lambda \sim \tilde{\mathcal{D}}_\lambda}\left[(1-\lambda)^2 \varphi(1-\lambda)\right],$$

*where $\tilde{\mathcal{D}}_\lambda$ is a uniform mixture of two Beta distributions, i.e., $\frac{\alpha}{\alpha+\beta} Beta(\alpha+1, \beta) + \frac{\beta}{\alpha+\beta} Beta(\beta + 1, \alpha)$ and $\psi(\cdot)$ is some function with $\lim_{a \to 0} \psi(a) = 0$. Moreover,*

$$\mathcal{R}_1(\theta, S) = \frac{\mathbb{E}_{\lambda \sim \tilde{\mathcal{D}}_\lambda}[1-\lambda]}{n} \sum_{i=1}^{n} w_i \left(h'(f_\theta(x_i)) - y_i\right) \nabla f_\theta(x_i)^\top \mathbb{E}_{r_x \sim \mathcal{D}_X} [r_x - x_i]$$

$$\mathcal{R}_2(\theta, S) = \frac{\mathbb{E}_{\lambda \sim \tilde{\mathcal{D}}_\lambda}\left[(1-\lambda)^2\right]}{2n} \sum_{i=1}^{n} w_i h''(f_\theta(x_i)) \nabla f_\theta(x_i)^\top \mathbb{E}_{r_x \sim \mathcal{D}_X}\left[(r_x - x_i)(r_x - x_i)^\top\right] \nabla f_\theta(x_i)$$

$$\mathcal{R}_3(\theta, S) = \frac{\mathbb{E}_{\lambda \sim \tilde{\mathcal{D}}_\lambda}\left[(1-\lambda)^2\right]}{2n} \sum_{i=1}^{n} w_i \left(h'(f_\theta(x_i)) - y_i\right) \mathbb{E}_{r_x \sim \mathcal{D}_X}\left[(r_x - x_i) \nabla^2 f_\theta(x_i)(r_x - x_i)^\top\right].$$

*Proof.* The corresponding mixup version is

$$
\begin{aligned}
L_n^{\text{mix}}(\theta, S) &= \frac{1}{n^2}\mathbb{E}_{\lambda \sim Beta(\alpha,\beta)}\sum_{i,j=1}^{n}[\lambda w_i h(f_\theta(\tilde{x}_{i,j}(\lambda))) - \lambda w_i y_i \\
&\qquad\qquad\qquad + (1-\lambda)w_j h(f_\theta(\tilde{x}_{i,j}(\lambda))) - (1-\lambda)w_j y_j] \\
&= \frac{1}{n^2}\mathbb{E}_{\lambda \sim Beta(\alpha,\beta)}\mathbb{E}_{B \sim Bern(\lambda)}\sum_{i,j=1}^{n}[w_i B(h(f_\theta(\tilde{x}_{i,j})) - y_i) \\
&\qquad\qquad\qquad + w_j(1-B)(h(f_\theta(\tilde{x}_{i,j})) - y_j)] \\
&= \frac{1}{n^2}\sum_{i,j=1}^{n}\{\frac{\alpha}{\alpha+\beta}\mathbb{E}_{\lambda \sim Beta(\alpha+1,\beta)}w_i[h(f_\theta(\tilde{x}_{i,j})) - y_i] \\
&\qquad\qquad\qquad + \frac{\beta}{\alpha+\beta}\mathbb{E}_{\lambda \sim Beta(\alpha,\beta+1)}w_j[h(f_\theta(\tilde{x}_{i,j})) - y_j])\} \\
&= \frac{1}{n}\sum_{i=1}^{n}w_i\mathbb{E}_{\lambda \sim \tilde{D}_\lambda}\mathbb{E}_{r_x \sim D_x^w}h(f(\theta, \lambda x_i + (1-\lambda)r_x)) - y_i f(\theta, \lambda x_i + (1-\lambda)r_x) \\
&= \frac{1}{n}\sum_{i=1}^{n}w_i\mathbb{E}_{\lambda \sim \tilde{D}_x}l_{\check{x}_i,y_i}(\theta),
\end{aligned}
$$

where $D_x^w = \frac{1}{n}\sum_{i=1}^{n}w_i\delta_i$ and $\check{x}_i = \lambda x_i + (1-\lambda)r_x$.

We let $\alpha = 1 - \lambda$ and $\psi_i(\alpha) = l_{\check{x}_i,y_i}(\theta)$. Then since we know $\psi_i$ is twice-differential, we have

$$
l_{\check{x}_i,y_i}(\theta) = \psi_i(\alpha) = \psi_i(0) + \psi_i'(0)\alpha + \frac{1}{2}\psi_i''(0)\alpha^2 + \alpha^2\varphi_i(\alpha).
$$

By the proof of Lemma 3.1 in [76] we know

$$
\begin{aligned}
\psi_i'(0) &= (h'(f_\theta(x_i)) - y_i)\nabla f_\theta(x_i)^\top (r_x - x_i), \\
\psi_i''(0) &= h''(f_\theta(x_i))\nabla f_\theta(x_i)^\top (r_x - x_i)(r_x - x_i)^\top \nabla f_\theta(x_i) \\
&\qquad + (h'(f_\theta(x_i)) - y_i)(r_x - x_i)^\top \nabla^2 f_\theta(x_i)(r_x - x_i).
\end{aligned}
$$

$\square$

**Lemma A.2.** *Consider the centralized dataset, i.e., $\frac{1}{n}\sum_{i=1}^{n}x_i = 0$, we have*

$$
\mathbb{E}_{\lambda \sim \tilde{D}_\lambda}[L_n^{mix}(\theta, \tilde{S})] \approx L_n^{std}(\theta, S) + \frac{1}{2n}[\sum_{i=1}^{n}w_i A''(x_i^\top \theta)]\mathbb{E}_{\lambda \sim \tilde{D}_\lambda}(\frac{(1-\lambda)^2}{\lambda^2})\theta^\top \widehat{\Sigma}_X \theta,
$$

*where $\widehat{\Sigma}_X = \frac{1}{n}\sum_{i=1}^{n}w_i x_i x_i^\top$, and the expectation is taken with respect to the randomness of $\lambda$.*

*Proof.* For GLM, the prediction is invariant to the scaling of the training data and thus we consider the re-scaled dataset $\tilde{S} = \{(\tilde{x}_i, y_i)\}_{i=1}^{n}$ where $\tilde{x}_i = \frac{1}{\lambda}(\lambda x_i + (1-\lambda)r_x)$. For GLM the mixed stadard loss function is

$$
L_n^{std}(\theta, \tilde{S}) = \frac{1}{n}\sum_{i=1}^{n}w_i l_{\tilde{x}_i,y_i}(\theta) = \frac{1}{n}\sum_{i=1}^{n}-w_i(y_i \tilde{x}_i^\top \theta - A(\tilde{x}_i^\top \theta)).
$$

In the proof of Lemma 3.3 in [76], we know by taking expectation with respect to the randomness of $\lambda$ and $r_x$ we have the following second-order approximation for the GLM loss,

$$
\mathbb{E}[L_n^{std}(\theta, \tilde{S})] \approx L_n^{std}(\theta, S) + \frac{1}{2n}[\sum_{i=1}^{n}w_i A''(x_i^\top \theta)]\mathbb{E}(\frac{(1-\lambda)^2}{\lambda^2})\theta^\top \widehat{\Sigma}_X \theta,
$$

where $\widehat{\Sigma}_X = \frac{1}{n}\sum_{i=1}^{n}w_i x_i x_i^\top$. $\square$

**Lemma A.3.** *Assume that the distribution of $x_i$ is $\rho$-retentive, i.e., satisfies the Assumption 5.1. Then the empirical Rademacher complexity of $\mathcal{W}_r$ satisfies*

$$Rad(\mathcal{W}_r, \mathcal{S}) \leq \max\{(\frac{\gamma(\delta)}{\rho})^{1/4}, (\frac{\gamma(\delta)}{\rho})^{1/2}\} \cdot \sqrt{\frac{rank(\Sigma_X)}{n}},$$

*with probability at least $1 - \delta$ for some constant $\gamma(\delta)$ that only depends on $\delta$.*

*Proof.* The proof is mainly based on [76]. By definition, given $n$ i.i.d. Rademacher rv. $\xi_1, \ldots, \xi_n$, the empirical Rademacher complexity is

$$\mathrm{Rad}\,(\mathcal{W}_\gamma, S) = \mathbb{E}_\xi \sup_{a(\theta) \cdot \theta^\top \Sigma_X \theta \leq \gamma} \frac{1}{n} \sum_{i=1}^{n} \xi_i \theta^\top x_i$$

Let $\tilde{x}_i = \Sigma_X^{\dagger/2} x_i$, $a(\theta) = \mathbb{E}_x\left[A''\left(x^\top \theta\right)\right]$ and $v = \Sigma_X^{1/2}\theta$, then $\rho$-retentiveness condition implies $a(\theta)^2 \geq \rho \cdot \min\left\{1, \mathbb{E}_x\left(\theta^\top x\right)^2\right\} \geq \rho \cdot \min\left\{1, \theta^\top \Sigma_X \theta\right\}$ and therefore $a(\theta) \cdot \theta^\top \Sigma_X \theta \leq \gamma$ implies that $\|v\|^2 = \theta^\top \Sigma_X \theta \leq \max\left\{\left(\frac{\gamma}{\rho}\right)^{1/2}, \frac{\gamma}{\rho}\right\}$.

As a result,

$$\mathrm{Rad}\,(\mathcal{W}_\gamma, S) = \mathbb{E}_\xi \sup_{a(\theta) \cdot \theta^\top \Sigma_X \theta \leq \gamma} \frac{1}{n} \sum_{i=1}^{n} \xi_i \theta^\top x_i$$

$$= \mathbb{E}_\xi \sup_{a(\theta) \cdot \theta^\top \Sigma_X \theta \leq \gamma} \frac{1}{n} \sum_{i=1}^{n} \xi_i v^\top \tilde{x}_i$$

$$\leq \mathbb{E}_\xi \sup_{\|v\|^2 \leq \left(\frac{\gamma}{\rho}\right)^{1/2} \vee \frac{\gamma}{\rho}} \frac{1}{n} \sum_{i=1}^{n} \xi_i v^\top \tilde{x}_i$$

$$\leq \frac{1}{n} \cdot \left(\frac{\gamma}{\rho}\right)^{1/4} \vee \left(\frac{\gamma}{\rho}\right)^{1/2} \cdot \mathbb{E}_\xi \left\|\sum_{i=1}^{n} \xi_i \tilde{x}_i\right\|$$

$$\leq \frac{1}{n} \cdot \left(\frac{\gamma}{\rho}\right)^{1/4} \vee \left(\frac{\gamma}{\rho}\right)^{1/2} \cdot \sqrt{\mathbb{E}_\xi \left\|\sum_{i=1}^{n} \xi_i \tilde{x}_i\right\|^2}$$

$$\leq \frac{1}{n} \cdot \left(\frac{\gamma}{\rho}\right)^{1/4} \vee \left(\frac{\gamma}{\rho}\right)^{1/2} \cdot \sqrt{\sum_{i=1}^{n} \tilde{x}_i^\top \tilde{x}_i}$$

Consequently,

$$\mathrm{Rad}\,(\mathcal{W}_\gamma, S) = \mathbb{E}_S\left[\mathrm{Rad}\,(\mathcal{W}_\gamma, S)\right] \leq \frac{1}{n} \cdot \left(\frac{\gamma}{\rho}\right)^{1/4} \vee \left(\frac{\gamma}{\rho}\right)^{1/2} \cdot \sqrt{\sum_{i=1}^{n} \mathbb{E}_{x_i}\left[\tilde{x}_i^\top \tilde{x}_i\right]}$$

$$\leq \frac{1}{\sqrt{n}} \cdot \left(\frac{\gamma}{\rho}\right)^{1/4} \vee \left(\frac{\gamma}{\rho}\right)^{1/2} \cdot \mathrm{rank}\,(\Sigma_X)$$

Based on this bound on Rademacher complexity, Theorem 5.1 can be proved by directly applying the Theorem 8 from [7]. □

# B   Experimental details

In this section, we present experimental setup in detail. Specifically, we describe the backbone model for each dataset in Sec. B.1, the detailed datasets description in Sec. B.2, the implementation details in Sec. B.3, uncertainty quantification results on simulated dataset in Sec. B.4, training accuracy of different subpopulations throughout training process in Sec. B.5 and additional results in Sec. B.6.

## B.1 Backbone model

Within each dataset, we keep the same model architecture as in previous work [70]: ResNet-50 [23] for Waterbirds and CelebA, DistilBERT [16] for CivilComments, and DenseNet-121 for Camelyon17. For ResNet-50, we used the PyTorch [55] implementation pre-trained with ImageNet. For DistilBERT, we employ the HuggingFace [67] implementation and start from the pre-trained weights. Same as previous work [70], for DenseNet-121 we employ the implementation without pretraining.

## B.2 Datasets details

We describe the datasets used in the experiments in detail and summarize the datasets in Table 4.

- **WaterBirds** [58]. The task of this dataset is to distinguish whether the bird is a waterbird or a landbird. According to the background and label of an image, this dataset has four predefined subpopulations, i.e., "landbirds on land", "landbirds on water", "waterbirds on land" , and "waterbirds on water". In the training set, the largest subpopulation is "landbirds on land" with 3,498 samples, while the smallest subpopulation is "landbirds on water" with only 56 samples.

- **CelebA** [43]. CelebA is a well-known large-scale face dataset. Same as previous works [41, 58], we employ this dataset to predict the color of the human hair as "blond" or "not blond". There are four predefined subpopulations based on gender and hair color, i.e., "dark hair, female", "dark hair, male", "blond hair, female" and "blond hair, male" with 71,629, 66,874, 22,880, and 1,387 training samples respectively.

- **CivilComments** [9]. For this dataset, the task is to classify whether an online comment is toxic or not, where according to the demographic identities (e.g., Female, Male, and White) and labels, 16 overlapping subpopulations can be defined. We use 269,038, 45,180, and 133,782 samples as training, validation, and test datasets respectively.

- **Camelyon17** [5, 33]. Camelyon17 is a pathological image dataset with over 450, 000 lymph-node scans used to distinguish whether there is cancer tissue in a patch. The training data is drawn from three hospitals, while the validation and test data are sampled from other hospitals. However, due to the different coloring methods, even the same hospital samples have different distributions. Therefore, we cannot get reliable subpopulation labels of Camelyon17.

Table 4: Summary of the datasets used in the experiments.

| Datasets | Labels | Groups | Population type | Data type | Backbone model |
|----------|--------|--------|-----------------|-----------|----------------|
| Waterbirds | 2 | 2 | Label×Group | Image | ResNet-50 |
| CelebA | 2 | 2 | Label×Group | Image | ResNet-50 |
| CivilComments | 2 | 8 | Label×Group | Text | DistilBERT-uncased |
| Camelyon17 | 2 | 5 | Group | Image | DenseNet-121 |

## B.3 Implementation details

In this section, we present the implementation details of all approaches. We implement our method in the codestack released with the WILDS datasets [33]. For some comparative methods, including ERM, IRM [3], IB-IRM [1], V-REx [34], CORAL [63], Group DRO [58], DomainMix [69], Fish [60], LISA [70], vanilla mixup and in-group mixup, we directly use the results in previous work [70]. For JTT [41], on the Waterbirds and CelebA datasets, we directly report the results in the paper, and on the CivilComments dataset, due to a different backbone model being employed, we reimplement the algorithm for fairly comparison. Same as the proposed method, we reimplement other methods in the codestack released with the WILDs datasets. We employ vanilla mixup on WaterBirds and Camelyon17 datasets. On CelebA and CivilComments datasets, we employ cutmix [71] and manifoldmix [65] respectively. For all approaches, we tune all hyperparameters with AutoML toolkit NNI [49] based on validation performance. Then we run the experiment multiple times on a computer with 8 Tesla V100 GPUs with different seeds to obtain the average performance and standard deviation. The selected hyperparameters for Algorithm 1 and Algorithm 2 are listed in Tabel 5.

Table 5: Hyperparameter settings for Algorithm 1 and Algorithm 2.

(a) Hyperparameter settings for Algorithm 1.

|  | WaterBirds | CelebA | CivilComments | Camelyon17 |
|---|---|---|---|---|
| Learning rate | 1e-5 | 1e-4 | 5e-5 | 1e-5 |
| Weight decay | 1 | 1e-4 | 1e-4 | 1e-2 |
| Batch size | 64 | 128 | 128 | 32 |
| Optimizer | SGD | SGD | AdamW | SGD |
| Hyperparameter $\alpha$ | 0.5 | 1.5 | 0.5 | 0.5 |
| Hyperparameter $\sigma$ | 0.5 | 0.5 | 1 | 1 |
| Maximum Epoch | 300 | 20 | 10 | 5 |

(b) Hyperparameter settings for Algorithm 2.

|  | WaterBirds | CelebA | CivilComments | Camelyon17 |
|---|---|---|---|---|
| Learning rate | 1e-5 | 1e-5 | 1e-05 | 1e-3 |
| Weight decay | 1 | 1e-1 | 1e-2 | 1e-2 |
| Batch size | 64 | 128 | 128 | 32 |
| Optimizer | SGD | SGD | AdamW | SGD |
| Start epoch $T_s$ | 50 | 0 | 0 | 0 |
| Sampling epoch $T$ | 50 | 5 | 5 | 5 |
| Hyperparameter $\eta$ | 80 | 50 | 3 | 5 |

## B.4 Uncertainty quantification results on simulated dataset

We conduct a toy experiment to show the uncertainty quantification could work well on the dataset with subpopulation shift. Specifically, we construct a four moons dataset (i.e., a dataset with four subpopulations) as shown in Fig. B.4. We compare our approximation (i.e., Eq. 6) with the following ensemble-based approximation:

$$u_i \approx \frac{1}{T} \sum_{t=1}^{T} \kappa(y_i, \hat{f}_{\theta_t}(x_i)) p(\theta_t; \mathcal{D}) d\theta. \tag{8}$$

Specifically, we train $T$ models and then ensemble them. The quantification results are shown in Fig. 3. We can observe that (1) the proposed historical-based uncertainty quantification method could work well on the simulated dataset; (2) compared with the ensemble-based method, the proposed method could better characterize the subpopulation shift.

## B.5 Training accuracy throughout training

We present how the training accuracy change throughout training in Fig. 4 on the CelebA and Waterbirds datasets to empirically show why the proposed estimation approach could work. From the experimental results, we observe that during training, easy groups with sufficient samples can be fitted well, and vice versa. For example, on the CelebA dataset, Group 0 and Group 1 with about 72K and 67K training samples quickly achieved over 95% accuracy. The accuracy rate on Group 2, which has about 23K training samples, increased more slowly and finally reached around 84%. The accuracy on Group 3, which has only about 1K training samples, is the lowest. Meanwhile, On the Waterbirds dataset, the samples of hard-to-classify group (e.g., Group 1) are also more likely to be forgotten by the neural networks.

## B.6 Additional results

In this section, we present the full results with standard deviation in Table 6, Table 7, and Table 8.

Table 6: Full comparison results with other methods in the group-oblivious setting where NA indicates the standard deviation in the original paper [41] is not available. The best results are in bold blue.

| | Waterbirds | | CelebA | |
| | Avg. | Worst | Avg. | Worst |
|---|---|---|---|---|
| ERM | **97.0 ± 0.2%** | 63.7 ± 1.9% | **94.9 ± 0.2%** | 47.8 ± 3.7% |
| Focal Loss [40] | 87.0 ± 0.5% | 73.1 ± 1.0% | 88.4 ± 0.3% | 72.1 ± 3.8% |
| CVaR-DRO [38] | 90.3 ± 1.2% | 77.2 ± 2.2% | 86.8 ± 0.7% | 76.9 ± 3.1% |
| CVaR-DORO [72] | 91.5 ± 0.7% | 77.0 ± 2.8% | 89.6 ± 0.4% | 75.6 ± 4.2% |
| $\chi^2$-DRO [38] | 88.8 ± 1.5% | 74.0 ± 1.8% | 87.7 ± 0.3% | 78.4 ± 3.4% |
| $\chi^2$-DORO [72] | 89.5 ± 2.0% | 76.0 ± 3.1% | 87.0 ± 0.6% | 75.6 ± 3.4% |
| JTT [41] | 93.6 ± (NA)% | 86.0 ± (NA)% | 88.0 ± (NA)% | 81.1 ± (NA)% |
| Ours | 93.0 ± 0.5% | **90.0 ± 1.1%** | 90.1 ± 0.4% | **85.3 ± 4.1%** |

| | CivilComments | | Camelyon17 |
| | Avg. | Worst | Avg. |
|---|---|---|---|
| ERM | **92.2 ± 0.1%** | 56.0 ± 3.6% | 70.3 ± 6.4% |
| Focal Loss [40] | 91.2 ± 0.5% | 60.1 ± 0.7% | 68.1 ± 4.8% |
| CVaR-DRO [38] | 89.1 ± 0.4% | 62.3 ± 0.7% | 70.5 ± 5.1% |
| CVaR-DORO [72] | 90.0 ± 0.4% | 64.1 ± 1.4% | 67.3 ± 7.2% |
| $\chi^2$-DRO [38] | 89.4 ± 0.7% | 64.2 ± 1.3% | 68.0 ± 6.7% |
| $\chi^2$-DORO [72] | 90.1 ± 0.5% | 63.8 ± 0.8% | 68.0 ± 7.5% |
| JTT [41] | 90.7 ± 0.3% | 67.4 ± 0.5% | 69.1 ± 6.4% |
| Ours | 90.6 ± 0.4% | **70.1 ± 0.9%** | **75.1 ± 5.9%** |

Table 7: Full comparison results with the algorithms **using training group labels** (Our method does not depend on this type of information). Results of baseline models are from [70]. The best three results are in bold brown or bold blue and the color indicates whether the train group label is used.

| | Group labels in train set? | Waterbirds | | CelebA | |
| | | Avg. | Worst | Avg. | Worst |
|---|---|---|---|---|---|
| IRM | Yes | 87.5 ± 0.7% | 75.6 ± 3.1% | **94.0 ± 0.4%** | 77.8 ± 3.9% |
| IB-IRM | Yes | 88.5 ± 0.6% | 76.5 ± 1.2% | **93.6 ± 0.3%** | 85.0 ± 1.8% |
| V-REx | Yes | 88.0 ± 1.0% | 73.6 ± 0.2% | 92.2 ± 0.1% | **86.7 ± 1.0%** |
| CORAL | Yes | 90.3 ± 1.1% | 79.8 ± 1.8% | **93.8 ± 0.3%** | 76.9 ± 3.6% |
| GroupDRO | Yes | **91.8 ± 0.3%** | **90.6 ± 1.1%** | 92.1 ± 0.4% | **87.2 ± 1.6%** |
| DomainMix | Yes | 76.4 ± 0.3% | 53.0 ± 1.3% | 93.4 ± 0.1% | 65.6 ± 1.7% |
| Fish | Yes | 85.6 ± 0.4% | 64.0 ± 0.3% | 93.1 ± 0.3% | 61.2 ± 2.5% |
| LISA | Yes | **91.8 ± 0.3%** | **89.2 ± 0.6%** | 92.4 ± 0.4% | **89.3 ± 1.1%** |
| Ours | No | **93.0 ± 0.5%** | **90.0 ± 1.1%** | 90.1 ± 0.4% | 85.3 ± 4.1% |

| | Group labels in train set? | CivilComments | | Camelyon17 |
| | | Avg. | Worst | Avg. |
|---|---|---|---|---|
| IRM | Yes | 88.8 ± 0.7% | 66.3 ± 2.1% | 64.2 ± 8.1% |
| IB-IRM | Yes | 89.1 ± 0.3% | 65.3 ± 1.5% | 68.9 ± 6.1% |
| V-REx | Yes | **90.2 ± 0.3%** | 64.9 ± 1.2% | 71.5 ± 8.3% |
| CORAL | Yes | 88.7 ± 0.5% | 65.6 ± 1.3% | 59.5 ± 7.7% |
| GroupDRO | Yes | 89.9 ± 0.5% | 70.0 ± 2.0% | 68.4 ± 7.3% |
| DomainMix | Yes | **90.9 ± 0.4%** | 63.6 ± 2.5% | 69.7 ± 5.5% |
| Fish | Yes | 89.8 ± 0.4% | **71.1 ± 0.4%** | **74.7 ± 7.1%** |
| LISA | Yes | 89.2 ± 0.9% | **72.6 ± 0.1%** | **77.1 ± 6.5%** |
| Ours | No | **90.6 ± 0.5%** | **70.1 ± 0.9%** | **75.1 ± 5.9%** |

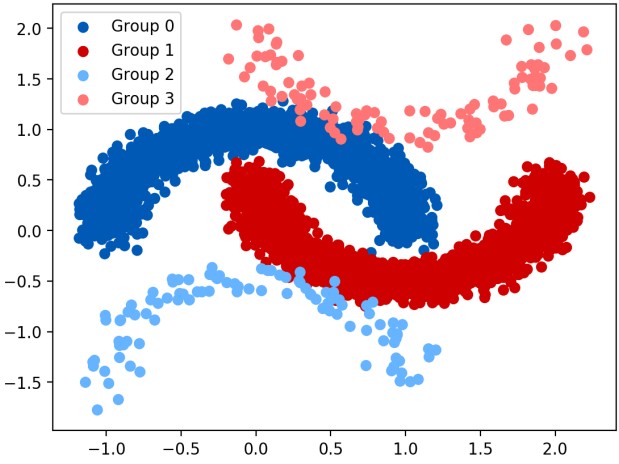

Figure 2: Simulated dataset with four different subpopulations. In the four subpopulations, Group 0 and Group 2 have the same label and groups 1 and 3 have the same labels.

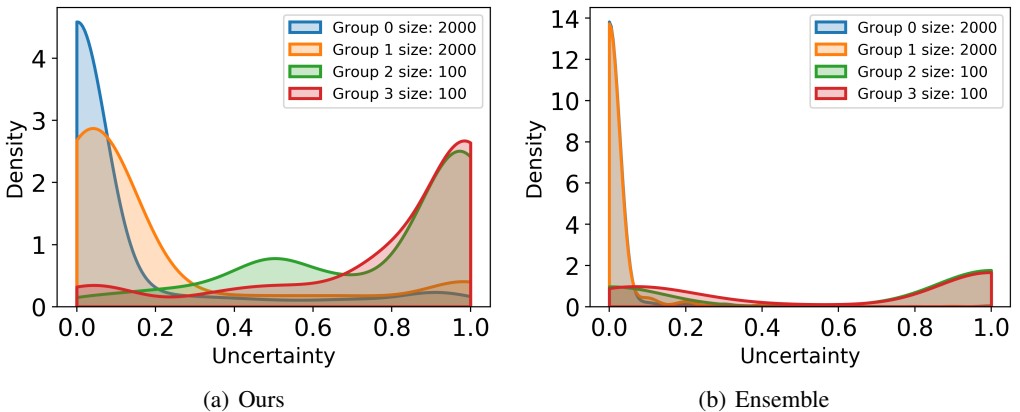

Figure 3: Visualization of the obtained uncertainty with kernel density estimation on simulated dataset, where group size refers to the sample number of the group.

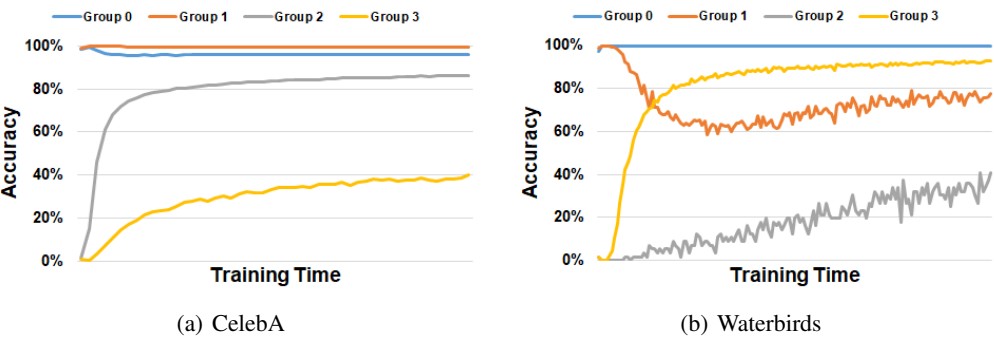

Figure 4: Visualization of the changing of training accuracy on different groups of CelebA and Waterbirds datasets.

Table 8: Full comparison with ERM and mixup based methods. Results of baseline models are from [70]. The best results are in bold brown or bold blue and the color indicates whether the train group label is used.

| | Group labels in train set? | Waterbirds Avg. | Worst | CelebA Avg. | Worst |
|---|---|---|---|---|---|
| ERM | No | **97.0 ± 0.2%** | 63.7 ± 1.9% | 94.9 ± 0.2% | 47.8 ± 3.7% |
| vanilla mixup | No | 81.0 ± 0.2% | 56.2 ± 0.2% | **95.8 ± 0.0%** | 46.4 ± 0.5% |
| in-group mixup | Yes | 88.7 ± 0.3% | 68.0 ± 0.4% | 95.2 ± 0.3% | 58.3 ± 0.9% |
| Ours | No | 93.0 ± 0.5% | **90.0 ± 1.1%** | 90.1 ± 0.4% | **85.3 ± 4.1%** |

| | Group labels in train set? | CivilComments Avg. | Worst | Camelyon17 Avg. |
|---|---|---|---|---|
| ERM | No | **92.2 ± 0.1%** | 56.0 ± 3.6% | 70.3 ± 6.4% |
| vanilla mixup | No | 90.8 ± 0.8% | 67.2 ± 1.2% | 71.2 ± 5.3% |
| in-group mixup | Yes | 90.8 ± 0.6% | 69.2 ± 0.8% | **75.5 ± 6.7%** |
| Ours | No | 90.6 ± 0.5% | **70.1 ± 0.9%** | 75.1 ± 5.9% |

## C   Justification for choosing historical-based uncertainty score

We employ the information from the historical training trajectory to approximate the sampling process because it is simple and effective in practice. Empirically, in contrast to other typical uncertainty quantification methods such as Bayesian learning or model ensemble [17, 36], our method can significantly reduce the computational and memory-storage cost by employing the information from the historical training trajectory, since Bayesian learning or model ensemble needs to sample/save multiple DNN models and performs inference computations on them. Meanwhile, our method has achieved quite promising final accuracy in contrast to other methods. In summary, we choose an uncertainty score that can achieve satisfactory performance while being more memory and computationally efficient.

## D   Societal impact and limitations

### D.1   Societal impact

Algorithmic fairness and justice are closely related to our work. Philosophically, there are two different views on justice. Firstly, Jeremy Bentham believes "the greatest good for the greatest number" can be seen as justice [50]. ERM can be considered to inherit this spirit which pays more attention to minimizing the majority subpopulation risks. Different from Jeremy Bentham's opinion, Rawlsian distributive justice [57] argues that we should maximize the welfare of the worst-off group. The proposed method and other IW-based methods can be seen as the practice of Rawlsian distributive justice due to focusing more on the minority subpopulations. However, in practice, the proposed method and other IW-based methods may sacrifice the average accuracy. Therefore, the ones using the proposed method need to carefully consider what fairness and justice are in a social context to decide whether to sacrifice the average accuracy and improve the worst-case accuracy.

### D.2   Limitations and future works

Even though the proposed method achieves excellent performance, it still has some potential limitations. (1) Similar to other IW-based methods, the proposed method may sacrifice the average accuracy. Therefore, it is also important and valuable to conduct a theoretical analysis of this phenomenon and explore novel ways to improve the worst-case accuracy of the model without sacrificing the average accuracy in the future work. (2) Although our method does not require training set group labels, how to leverage unreliable subpopulation information (e.g., subpopulation labels are noise) to improve UMIX would be a promising research topic. For example, when the unreliable subpopulation labels are available, UMIX could be improved by equipping with existing importance weighting methods. (3) Similar to the previous IW-based methods, the label noise is also not considered in our method,

which may lead to over-focusing on noisy samples. Currently, it's still a challenging open problem to distinguish the minority samples from the mislabeled noise samples in the data with subpopulation shift. (4) At the same time, this work only considers subpopulation shifts on Euclidean data, hence it is also a promising future direction to generalize IW-based methods to graph-structured data, under the guidance of invariance principle on graphs, such as that of [12]. We leave them as important future works.