# OpenReview forum: "UMIX: Improving Importance Weighting for Subpopulation Shift via Uncertainty-Aware Mixup"
_NeurIPS.cc/2022/Conference — NeurIPS 2022 Accept_

### Official Review · Reviewer_Vm21 · 2022-07-06

**Rating:** 7
**Confidence:** 3
**Soundness:** 3 good
**Presentation:** 3 good
**Contribution:** 3 good

**Summary:**

The authors propose a importance reweighting scheme for mixup which promotes a fair training balance between subgroups in the data. The does not require group
labels during training.

**Questions:**

- Section 5 states "Thus our bound can be tighter when
315 the intrinsic dimension of data is small (i.e., rank(Σ) ≪ d)." Is this really useful? Can you point to some intuition or concrete examples of when this would usefully lower the bound? When can we reasonably expect that to not be full rank? On the same note, is this referring to the dimension in the input space, the output space before the loss function, or something else entirely?

- Are you actually MC sampling the saved SWAG models by building the Gaussian
posterior throughout the training process in order to get the importance
weights? OR are you just using the snapshots to get the model disagreement?

- Section 3.1 says that the $N$ training examples are i.i.d. and sampled from
$P$, but then immediately says that the subpopulation $g$ follows the
distribution $P_g$. Shouldn't the assumption then be that the dataset is not
i.i.d. and only conditionally i.i.d. given $P_g$?

**Limitations:**

The authors have addressed both in the appendix.

**Strengths And Weaknesses:**

# Strengths

- The paper is well placed among recent works in both mixup, and importance reweighting methods.
- The method performs well without group information during training which is impressive
- The theoretical analysis provides a nice
- Overall I think this is a solid work, but I some clarification on the issues raised in the following sections.

# Weaknesses

- L190 says that you assume a reasonable importance weight is linearly
positively correlated to the corresponding uncertainty. What is a 'reasonable'
importance weight, and on what basis can this assumption be made? I realize
this is shown in Figure 1, but there should be some provided justification or
reference to the fact that this will be empirically tested later.

- How does UMIX perform without the group labels in the validation set? If the
method is assumed to be oblivious to the group information, then it may not be
reasonable to assume that they are always in the validation set.

# Minor

- L113: BNN's is proposed --> BNN's have been proposed.
- L115: Performing ensembles --> ensembling them.
- L122: upweight mixed samples thus can encourage --> upweight mixed samples, and thus can encourage.
- L158: weights of UMIX are posed --> What does this mean?
- L178: $p(theta; \mathcal{D})$ --> shouldn't this be $p(\theta | \mathcal{D})$?
- L180: with $T$ Monte Carlo Sampling --> with $T$ Monte Carlo samples
- L193: in practice we could set $c = 1$ --> Does this mean that you do set $c=1$ in practice? It should be clearly stated if that is the case.
- L295: proportion is --> proportion being
- L296: in specific --> specifically

---

> ### Author Response · Authors · 2022-08-02
> **Response to reviewer Vm21  (part 2/2)**
>
> ***Q4. Are you actually MC sampling the saved SWAG models by building the Gaussian posterior throughout the training process in order to get the importance weights? OR are you just using the snapshots to get the model disagreement?***
>
> R4. Unlike SWAG, we didn't build the Gaussian posterior, since our method aims to obtain the uncertainty of the training samples. We employ a more efficient way to avoid the high computational and memory-storage overhead during sampling from the model posterior distribution. Specifically, compared with SWAG or other uncertainty estimation methods, our method is more efficient from two perspectives. First, we do not need to estimate the posterior distribution of the model parameters. Second, we do not need to sample lots of models from the estimated posterior distribution again to obtain uncertainty. Therefore, in practice, our model is much more efficient compared SWAG.
>
> ***Q5. Section 3.1 says that the $N$ training examples are i.i.d. and sampled from $P$, but then immediately says that the subpopulation $g$ follows the distribution $P_g$. Shouldn't the assumption then be that the dataset is not i.i.d. and only conditionally i.i.d. given $P_g$?***
>
> R5. Thanks for the suggestion. For clarification, we will revise it to "The setting can be considered as that the training distribution $P$ to be a mixture of $G$ groups $P_g$." in the revision.
>
> ***Other minor concerns***
>
> We also thank the reviewers for their other detailed suggestions, which helped us a lot. We will revise the paper based on the comments.
>
> **Reference**
>
> [1] Zhai R, Dan C, Kolter Z, et al. Doro: Distributional and outlier robust optimization[C]//International Conference on Machine Learning. PMLR, 2021: 12345-12355.
>
> [2] Liu, Evan Z., et al. "Just train twice: Improving group robustness without training group information." International Conference on Machine Learning. PMLR, 2021.
>
> [3] Shen, Zheyan, et al. "Stable learning via sample reweighting." Proceedings of the AAAI Conference on Artificial Intelligence. Vol. 34. No. 04. 2020.

---

> > ### Comment · Reviewer_Vm21 · 2022-08-05
> > **Thanks**
> >
> > Thank you fro your explanation and responses.
> >
> > Not having the validation group labels cause a much smaller effect than I would have thought, which is promising. Overall I think this method is interesting and useful so I will adjust my score accordingly.

---

> ### Author Response · Authors · 2022-08-02
> **Response to reviewer Vm21 (part 1/2)**
>
>
> We thank the reviewer for the encouraging comments and insightful feedback. We address the concerns below.
>
> ***Q1. What is a 'reasonable' importance weight, and on what basis can this assumption be made?***
>
> R1. First, we assume the samples with high uncertainty can be given a higher weight and vice versa. This assumption is reasonable since samples in the majority group have low uncertainties with high probability as shown in Fig. 1. Holding this assumption, we chose a simple yet effective implementation, i.e., set importance weight that is linearly correlated to the uncertainty. As shown by our extensive results, this strategy can achieve satisfactory performance.
>
> ***Q2. How does UMIX perform without the group labels in the validation set?***
>
> R2. First, we added the evaluation on the Waterbirds and CelebA dataset without using the validation set group label information. Specifically, after each training epoch, we evaluate the performance of the current model on the validation set and save the model with the best average accuracy. Finally, we test the performance of the saved model on the testset. The experimental results are shown in the following tables. From the experimental results, we have the following observations: 1. When the validation set group information is not used, the worst-case ACC of our method drops a little while the average ACC improves a little. 2. Compared with JTT using validation set group information, our method still achieves excellent performance, such as on the Waterbirds dataset, which is still 2.9% higher than JTT in terms of worst-case accuracy.
>
> | Dataset    | Metric         | ERM   | JTT   | Ours  | Ours (without validation group label) |
> |------------|----------------|-------|-------|-------|---------------------------------------------|
> | Waterbirds | Average ACC    | 97.0% | 93.6% | 93.0% | 93.6%                                       |
> | Waterbirds | Worst-case ACC | 63.7% | 86.0% | 90.0% | 88.9%                                       |
> | CelebA     | Average ACC    | 94.0% | 88.0% | 90.1% | 90.4%                                       |
> | CelebA     | Worst-case ACC | 47.8% | 81.1% | 85.3% | 84.6%                                       |
>
> Second, we agree that it is an oracle strategy since it requires additional group information of validation set. However, as shown in some related works [1,2], the model selection with no group labels during validation is a very hard and ill-posed problem. Model selection always has a huge impact on the performance of the final model. Under our framework, this problem could be better solved. For example, we can estimate the uncertainty of the validation set samples and construct a pseudo-worst-case group based on the uncertainty of the validation set. Finally, we can reduce the impact of model selection by selecting the best model based on pseudo-worst-case accuracy.
>
> ***Q3. Section 5 states "Thus our bound can be tighter when the intrinsic dimension of data is small (i.e., rank($\Sigma$) $\ll d$)." Is this really useful? Can you point to some intuition or concrete examples of when this would usefully lower the bound? When can we reasonably expect that to not be full rank? On the same note, is this referring to the dimension in the input space, the output space before the loss function, or something else entirely?***
>
> R3. This is a very good question.
>
> ```
> "Is this referring to the dimension in the input space?"
> ```
> Yes, the intrinsic dimension of data refers to the input space's intrinsic dimension.
>
> ```
> "When can we reasonably expect that to not be full rank?"
> ```
> When the features are highly correlated, e.g., some variables are relevant to the existing ones, which is prevalent in practice, rank($\Sigma$) would be much smaller than the input space dimension.
>
> ```
> "Is this really useful? Can you point to some intuition or concrete examples of when this would usefully lower the bound?"
> ```
> Literature from statistical learning has shown that redundant features may do serious harm to the generalization capability of the ERM [3]. Instead, this issue can be alleviated by our proposed UMIX framework, which smartly introduces data augmentation by inherently imposing an implicit regularization. Specifically, the data augmentation takes the subpopulation shift into account by leveraging the knowledge from the heterogeneity between different subpopulations to improve the generalization. Therefore, our theoretical bound is exactly the analysis of the above intuitive consideration. We will add some concrete, synthetic examples in the revision.

---

### Official Review · Reviewer_cPPR · 2022-07-09

**Rating:** 6
**Confidence:** 4
**Soundness:** 3 good
**Presentation:** 4 excellent
**Contribution:** 3 good

**Summary:**

The authors tackle the problem of subpopulation shift without subpopulation labels. They propose a method called uncertainty-aware mixup (UMIX), which augments regular mixup with importance weights computed based on predictive uncertainty. The authors benchmark their method against a large set of group aware and group unaware baselines on standard datasets, finding that their method shows competitive worst-case accuracy. Finally, the authors provide theoretical insight into why their method outperforms traditional importance weighting-based methods.

**Questions:**

Please comment on points 2-6 from above.

**Limitations:**

The authors adequately address the limitations in the appendix.

**Strengths And Weaknesses:**

Strengths:
- The paper is generally well-written and easy to follow.
- The authors compare against a wide range of methods on the standard datasets, and report their results with standard deviations (in the appendix).
- The method performs competitively relative to the baselines.

Weaknesses:
1. Though the method is novel, is it only an incremental change from what is proposed in JTT and LISA.
2. The authors should provide more empirical justification for the uncertainty measure that they use. Though it is inspired by SWAG, it does seem quite different from SWAG. How does the measure of uncertainty compare with SWAG, and why can SWAG not be used directly in this method (by sampling from the posterior over weights)?
3. It seems that the uncertainty measure takes both epistemic and aleatoric uncertainty into account. Would it be more ideal to weight based on epistemic uncertainty only, as some groups may inherently have more error than others?
4. It would be interesting to explore how the uncertainty measure (Fig. 1) changes over time during the process of running Algorithm 2. Does this empirically support the intuition from Lines 195-198?
5. For Camelyon-17, I would suggest showing performance on the source domain test set along with the target domain results.
6. It would be ideal if the authors could test on a couple more domain generalization datasets, as this seems relatively unexplored (only Camelyon-17), and many of the baseline methods are targeted towards the domain generalization setting.
7. The authors should state the range of hyperparameters tuned, as well as the total tuning budget, in Appendix B3.

---

> ### Author Response · Authors · 2022-08-02
> **Response to reviewer cPPR (part 2/2)**
>
>
> ***Q4. How the uncertainty measure changes over time during the process?***
>
> R4. We show the empirical results on the CelebA dataset in the following table and other details will be plotted as line charts to add to the revision.  From the experimental results, we observe that during training, easy groups with sufficient samples can be fitted well in the earlier training stage. Conversely, the harder group is learned more slowly by the model.
>
> For example, on the CelebA dataset, Group-0 and Group-1 with about 72K and 67K training samples quickly achieved over 95% accuracy. The accuracy rate on Group-2, which has about 23K training samples, increased more slowly and finally reached around 84%. The accuracy on Group-3, which has only about 1K training samples, is the lowest.
>
> |     Training   epoch    |     Group-0   ACC    |     Group-1   ACC    |     Group-2   ACC    |     Group-3   ACC    |
> |:-----------------------:|:--------------------:|:--------------------:|:--------------------:|:--------------------:|
> |             1           |         99.6%        |         100.0%       |         15.7%        |          0.0%        |
> |             5           |         95.9%        |         99.7%        |         72.1%        |         14.8%        |
> |            10           |         95.9%        |         99.5%        |         78.9%        |         26.1%        |
> |            15           |         95.9%        |         99.5%        |         81.4%        |         30.4%        |
> |            30           |         96.1%        |         99.5%        |         84.3%        |         36.6%        |
>
> ***Q5. Performance on the source domain test set along with the target domain results on Camelyon17.***
>
> R5. We present the experimental results on the source domain of our method and some comparison methods in the following table. From the experimental results, we can see that our method is more competitive on source domain data compared to JTT.
>
> |     ERM      |     JTT      |     Ours     |
> |--------------|--------------|--------------|
> |     95.2%    |     92.5%    |     93.7%    |
>
> ***Q6. It would be ideal if the authors could test on a couple more domain generalization datasets, as this seems relatively unexplored (only Camelyon-17), and many of the baseline methods are targeted towards the domain generalization setting.***
>
> R6. Thanks for your suggestion. In this paper, we mainly focus on the task of the subpopulation shift rather than domain generalization. Although we have shown that our method can improve the model robustness against out-of-domain settings on the Camelyon-17 dataset, we think this is just a preliminary exploration to validate its effectiveness on the domain generalization tasks. In practice, our method could achieve leading performance by focusing on these difficult samples with higher uncertainty in the training set.  However, the task of domain generalization is more challenging than the sub-population shift and can be seen as an ill-posed problem [2]. We think that combining the invariant feature learning [3] with an importance-weighting-based mixup may be a promising way to handle the domain generalization task. Therefore, we leave this idea as a possible future direction.
>
> **Reference**
>
> [1] Hashimoto, Tatsunori, et al. "Fairness without demographics in repeated loss minimization." International Conference on Machine Learning. PMLR, 2018.
>
> [2] Koh P W, Sagawa S, Marklund H, et al. Wilds: A benchmark of in-the-wild distribution shifts[C]//International Conference on Machine Learning. PMLR, 2021: 5637-5664.
>
> [3] Arjovsky M, Bottou L, Gulrajani I, et al. Invariant risk minimization[J]. arXiv preprint arXiv:1907.02893, 2019.

---

> ### Author Response · Authors · 2022-08-02
> **Response to reviewer cPPR (part 1/2)**
>
> We thank the reviewer for their very thorough read of the paper and their extensive feedback. We will consider their many suggestions with great care. We only respond to the most salient points below, but we will follow up on the unresolved suggestion in the future.
>
> ***Q1. Comparison with existing works.***
>
> R1. In the paper (L68 to L82), we have briefly described the differences between JTT and LISA.  Here we explain it in more detail.
>
> ---
> **Comparison with JTT.** JTT obtains and employs importance weights which is quite different from our method.
> * First, the importance weights are obtained in different ways. The proposed algorithm introduces uncertainty-based importance weights constructed by sampling multiple prediction results from the historical training trajectory, while JTT obtains importance weights by building an error set from a single prediction result. Therefore JTT can be seen as a special case of our method.
> * Second, the importance weights are employed in different ways. JTT just reweights the training samples during training, while we introduce sample weights into the vanilla mixup strategy to improve the importance weighting.
>
>
> ---
> **Comparison with LISA.** The settings and motivations of LISA and the  proposed method are different from our paper.
> * First, LISA assumed that the group label is available during training, which may limit its application in scenarios where group labels are inaccessible (e.g., anonymous samples for privacy protection [1]). Our paper studies the group-oblivious setting, i.e., without dependence on the group information, which enjoys wider applicability.
> * Second, LISA aims to improve the generalization of the model by mixing samples from the same subpopulation or the same label, while our method tries to improve the traditional importance reweighting through uncertainty-aware mixup.
>
> Therefore, our work is orthogonal to LISA, i.e., we can use our weight-building strategy to improve LISA.
>
> ***Q2. Why can SWAG not be used directly in this method?***
>
> R2. Thanks for your suggestions. We will add the following discussions in revision. In the proposed method, we perform a prediction ensemble using the historical model snapshots for uncertainty quantification. This is for both efficiency, implementation, and accuracy reasons.
>
> * For efficiency, in contrast to SWAG, our method can significantly reduce the computational and memory-storage cost, since SWAG needs to sample multiple DNN models and performs inference computations on them.
> * For implementation, our uncertainty estimation method is agnostic to the optimization algorithm and backbone network. SWAG or some other methods may be limited to the optimization algorithm and backbone network. For example, an SGD optimizer is required in SWAG to estimate the posterior while other more suitable optimizers may be required when using some backbone networks (e.g., an AdamW optimizer is often required in a Bert-based backbone).
> * For accuracy, our method has achieved quite promising final accuracy in contrast to other methods.
>
> In summary, we choose an uncertainty score that can achieve satisfactory performance while being more memory and computationally efficient. Further, we highlight that the main contribution of our paper is not focusing on specific uncertainty measures and the SWAG  can be also naturally integrated into the UMIX framework.
>
> ***Q3. Would it be more ideal to weight based on epistemic uncertainty only?***
>
> R3. This is a very insightful question. In fact, in our proposed method, the samples are indeed weighted only based on epistemic uncertainty, which can be understood from the following two perspectives.
> * First, our uncertainty is obtained by sampling from the model on the training trajectory, which can be seen as sampling from the model posterior in a more efficient way.
> * Second, as discussed in the limitation (L109 in the appendix), we consider that the training samples do not contain the inherent noise (aleatoric uncertainty) since it is usually intractable to distinguish between noisy samples and minority samples from data with subpopulation shifts. We leave this problem for future work.
>
> We will add these discussions in the revision.

---

### Official Review · Reviewer_dN9V · 2022-07-10

**Rating:** 7
**Confidence:** 3
**Soundness:** 3 good
**Presentation:** 3 good
**Contribution:** 3 good

**Summary:**

The paper proposes a new mixup method (UMIX) based on uncertainty estimation and importance weighting for subpopulation shift. The authors also prove theoretically that UMIX can achieve better generalization error bounds over IW methods without mixup.

**Questions:**


1. While the idea proposed by the paper is straightforward, there are some missing details that are essential to justify the method’s effectiveness. For example, UMIX can be seen as a new data augmentation method/loss function, but how is the training procedure like? Is it similar as ERM, or is it adopted from some other algorithm. Adding this discussion would help to verify whether the comparisons are fair or not.

2. The choices of parameters are missing and including them (such as T_{s} and T in Algorithm 2) would be nice.

3. In Figure1, why is there hardly any samples with uncertainty from 0.3 to 0.8? Is it an effect of KDE, or is it due to the attributes of the datasets?

Minor questions:

1. Line59: contirbution $\rightarrow$ contributions

**Limitations:**

The authors have thouroughly discussed the limitations and social impacts of their work in the appendix.

**Strengths And Weaknesses:**

**Strengths**

Originality: Though the ideas of uncertainty estimation, importance weighting and mixup are not new, the combination of these ideas are novel and intriguing.

Quality: The authors proved empirically and theoretically that their method is better than other baselines. The results are rather solid.

Clarity: The paper is well-written and easy to follow.

Significance: The paper provides a new method for augmenting mixup via uncertainty.

**Weaknesses**

I have not identified major weaknesses of this paper. However, I do have some concerns and listed them in the “Questions” part below.

---

> ### Author Response · Authors · 2022-08-02
> **Response to reviewer dN9V**
>
>
> We thank the reviewer for the positive feedback and address the concerns point-by-point.
>
> ***Q1. Training procedure details.***
>
> R1. Thanks for your suggestions. The training procedure is the same as vanilla ERM except for the proposed UMIX augmentation method/loss function, making our model more simple and general. More specifically, as shown in Algorithm 1, we first obtain the training samples from the training dataset, then perform the proposed method to augment the training samples and calculate the loss, and finally update the model parameters with an optimization algorithm.  Moreover, the complete code has been released at https://anonymous.4open.science/r/UMIX-64ED to help readers to understand more training details.
>
> ***Q2. Choices of hyperparameters.***
>
> R2. Thank you for your suggestions. Limited by space, the detailed hyperparameters setting for Algorithm 1 and Algorithm 2 is in Appendix B.3. For example, on the Waterbirds dataset, $T_s$ and $T$ are 50 and 10, respectively. For better clarification, we will move some main settings of hyperparameters to the main text in the next version.
>
> ***Q3. Why is there hardly any samples with uncertainty from 0.3 to 0.8?***
>
> R3. The strict concentration around 0.3 and 0.8 is due to the attributes of the training datasets with imbalanced subpopulation distributions. Specifically, the majority groups tend to have smaller uncertainties (less than 0.3) due to sufficient training samples. On the other hand, minority groups often have higher uncertainties (greater than 0.8) due to less attention during training. Actually, it also shows the effectiveness of the employed uncertainty estimation strategy since the sharpness of uncertainty distribution is good.

---

> > ### Comment · Reviewer_dN9V · 2022-08-04
> > **Further Comment**
> >
> > The authors have properly addressed my concerns and questions. The code is also made publicly available, which I appreciate much.
> >
> > I went through the other reviewers' comments and corresponding replies. Overall, I regard this paper to be a nice work that can be impactful in this area. The combination of mixup and importance weighting is novel, insightful theoretical analysis is provided, and SOTA performance is achieved.
> >
> > Currently I will keep my score fixed. Although I would like this paper to be accepted, I am also willing to listen to the opinions of the other reviewers, especially reviewer ptj2, whose score is divergent from the other reviewers.

---

### Official Review · Reviewer_ptj2 · 2022-07-12

**Rating:** 5
**Confidence:** 4
**Soundness:** 3 good
**Presentation:** 3 good
**Contribution:** 3 good

**Summary:**

The paper extends the Mixup method by utilizing a dynamic uncertainty score as importance weight for Mixup samples. The uncertainty score (for a given x_i) is obtained by tracking the average prediction error of x_i through out the training trajectory, as an indication of how "hard" it is to learn this example.

The method is evaluated on 4 datasets with subpopulation shift. The proposed method outperforms others on "worst case" accuracy, but under-performs the baseline (ERM) on average accuracy.

**Questions:**

The uncertainty score u_i is updated every epoch, right? i.e. as the training epoch (T) progresses, you will get updated value of u_i(T). If so, please reflect that in Algorithm 1.

In Algorithm 2, you don't have to "save" all the predictions for every example at every epoch. Keeping a running average of $\kappa(y_i, f_{\theta_t}(x_i))$ over epoch t for each $x_i$ would be more memory efficient, right?

**Strengths And Weaknesses:**

The overall writing is clear and easy to follow. However, I feel the originality and significance of the proposed method is thin. It is a straightforward extension of Mixup by combining importance weight and uncertainty score. The paper didn't explore or discuss other choices of this "uncertainty score" as importance weight. There's a whole literature on influence score (how influential is an example to the learning process), which can also play a similar role here.

The evaluation results are ok but not impressive. I'm not convinced that a better "worst" accuracy but degraded avg accuracy is a significant contribution, given all the extra compute required by UMix. Perhaps other reviewers could convince me how this is important and relevant to the research community.

---

> ### Author Response · Authors · 2022-08-02
> **Response to reviewer ptj2 (part 2/2)**
>
> ***Q4. How to obtain the uncertainty $u_i$?***
>
> R4. As shown in the manuscript (Line 158-161 and Algorithm 1), the uncertainty score $u_i$ and importance weights are fixed as input for all the epochs during training in Algorithm 1. We restate the basic procedure of the proposed method here.
>
> Our method is mainly composed of two steps, including obtaining training importance weights and uncertainty-based mixup training.
> * Firstly, we employ Algorithm 2 to obtain the uncertainty-based training importance weights.
> * Then, the training importance weights are fixed as static weights to conduct uncertainty-based mixup training, i.e., Algorithm 1.
>
> ***Q5. A suggestion about more memory-efficient implementation.***
>
> R5. This is a very useful suggestion and we will add this modification in the revision. In the current version, it has little impact on the training procedure because it only takes about $N*T$ floating points number of extra memory costs to save these results, where $N$ and $T$ are the number of training samples and epochs, respectively.
>
> **Reference**
>
> [1] Byrd J, Lipton Z. What is the effect of importance weighting in deep learning?[C]//International Conference on Machine Learning. PMLR, 2019: 872-881.
>
> [2] Sagawa S, Koh P W, Hashimoto T B, et al. Distributionally Robust Neural Networks[C]//International Conference on Learning Representations. 2020.
>
> [3] Zhai R, Dan C, Kolter Z, et al. Understanding Why Generalized Reweighting Does Not Improve Over ERM[J]. arXiv preprint arXiv:2201.12293, 2022.
>
> [4] Sagawa S, Raghunathan A, Koh P W, et al. An investigation of why overparameterization exacerbates spurious correlations[C]//International Conference on Machine Learning. PMLR, 2020: 8346-8356.
>
> [5] Xu D, Ye Y, Ruan C. Understanding the role of importance weighting for deep learning[C]//International Conference on Learning Representations. 2021.
>
> [6] Hashimoto T, Srivastava M, Namkoong H, et al. Fairness without demographics in repeated loss minimization[C]//International Conference on Machine Learning. PMLR, 2018: 1929-1938.
>
> [7] Japkowicz N. The class imbalance problem: Significance and strategies[C]//Proc. of the Int’l Conf. on Artificial Intelligence. 2000, 56: 111-117.
>
> [8] Gal Y, Ghahramani Z. Dropout as a bayesian approximation: Representing model uncertainty in deep learning[C]//international conference on machine learning. PMLR, 2016: 1050-1059.
>
> [9] Lakshminarayanan B, Pritzel A, Blundell C. Simple and scalable predictive uncertainty estimation using deep ensembles[J]. Advances in neural information processing systems, 2017, 30.
>
> [10] Liu E Z, Haghgoo B, Chen A S, et al. Just train twice: Improving group robustness without training group information[C]//International Conference on Machine Learning. PMLR, 2021: 6781-6792.
>
> [11] Michel P, Hashimoto T, Neubig G. Distributionally Robust Models with Parametric Likelihood Ratios[C]//International Conference on Learning Representations. 2022.
>
> [12] Piratla V, Netrapalli P, Sarawagi S. Focus on the Common Good: Group Distributional Robustness Follows[C]//International Conference on Learning Representations. 2022.

---

> ### Author Response · Authors · 2022-08-02
> **Response to reviewer ptj2 (part 1/2)**
>
> We appreciate the reviewer's comments and efforts sincerely. We believe the following point-to-point response can address all the concerns and misunderstandings.
>
> ***Q1. Clarify the originality and significance.***
>
> R1. In short, the proposed method explores a novel and smart way to combine the superiorities of both mixup and importance-weighting strategies. We found the combination can bring both empirical improvements and theoretical insights.
>
> Specificifically, the originality, and significance of our method are four-fold.
>
> * First, from the method perspective, as pointed out by the introduction and prior theoretical analysis [1, 2, 3, 4, 5], previous importance-weighting-based methods suffer an intrinsic limitation that their weighting effects will diminish along with the training proceeds, especially when they are applied to over-parameterized NNs. In this paper, motivated by the mixup strategy which can always produce "reasonable" novel samples, we propose a novel and smart way to combine the superiorities of both mixup and IW strategies.  Given the high-level idea, we further instantiate it as a general learning framework with effective engineering implementations and the use of high-quality uncertainty quantification. We found the combination can bring both empirical improvements and theoretical insights as shown below. To the best of our knowledge, we are the first to explore the above idea and have successfully validated its effectiveness to tackle the issue of subpopulation shift.
>
> * Second, from the theoretical perspective, we provide insightful analysis that the employment of mixup can achieve a distribution-dependent generalization bound in the subpopulation shift setting and the bound can be tighter than the one achieved by conventional importance weighting methods. This interesting theoretical finding can motivate the research community to explore the usage of the mixup for obtaining a more explainable and tight generalization bound.
>
> * Third, from the empirical perspective, we have validated that the proposed UMIX achieves excellent performance in both group-oblivious and group-aware settings. Specifically, in the group-oblivious setting, our method achieves state-of-the-art performance in terms of worst-case accuracy on three datasets with subpopulation shifts. Further, we can still achieve competitive performance without using group information compared with group-aware algorithms.
>
> * At last, from the application side, given the broad usage of importance-weighting methods in distribution shift, fair machine learning, and long-tailed classification [6, 7], the proposed technique can be applied to a wide range of scenarios to improve the generalization ability of IW-based methods.
>
> ***Q2. Other choices of this "uncertainty score".***
>
> R2. Thanks for your suggestions and we will add the following discussions in revision. At a high level, we perform a prediction ensemble using the historical model snapshots for uncertainty quantification. This is for both efficiency and accuracy reasons.
>
> * For efficiency, in contrast to other typical uncertainty quantification methods such as Bayesian learning or model ensemble [8,9], our method can significantly reduce the computational and memory-storage cost by employing the information from the historical training trajectory, since Bayesian learning or model ensemble needs to sample/save multiple DNN models and performs inference computations on them.
>
> * For accuracy, our method has achieved quite promising final accuracy in contrast to other methods.
>
> In summary, we choose an uncertainty score that can achieve satisfactory performance while being more memory and computationally efficient.
>
> Further, we highlight that the main contribution of our paper is not focusing on specific uncertainty measures and other measures (e.g., Bayesian-based or ensemble-based methods) can also be naturally integrated into the UMIX framework.
>
> ***Q3. Significance of worst-case accuracy.***
>
> R3. On the one hand, the worst-case accuracy is a standard metric in this research community. As we can see, many recent papers [2,10,11,12] in this line also employ the worst-case accuracy as an evaluation metric.
>
> On the other hand, the trade-off between the average and worst-case accuracy is a well-known challenge [6]. Different scenarios and applications will lay different emphasis on these two measures. For example, in fairness-related applications, we should pay more attention to the performance of the minority groups to reduce the gap between the majority groups and ensure the fairness of the machine learning decision system.
>
> It's worth noting that even though the proposed method focus on worst-case accuracy, it also achieves competitive performance in terms of average accuracy compared with recent methods.

---

> ### Author Response · Authors · 2022-08-08
> **Message to reviewer**
>
> Dear Reviewer,
>
> We are wondering whether your concerns have been properly addressed.
> If you have further questions after reading the answers, it would be great to let us know.
>
> Best regards,
> The authors.

---

> ### Comment · Reviewer_ptj2 · 2022-08-09
> **Adjusted rating**
>
> Thanks for addressing my concerns. After reading the rebuttal, I have adjusted my rating accordingly.

---

### Meta-Review · Area_Chair_TVd3 · 2022-08-25

**Recommendation:** Accept
**Confidence:** Less certain

**Metareview:**

The reviewers unanimously agreed here that incorporating uncertainty scores as importance weights for mixup, and empirically the authors' method seems to lead to substantial quantitative performance improvements. I think the heuristic use of the model's parameter history to estimate uncertainty is reasonable. However, while I am recommending acceptance, the SAC and I feel that there are a few concerns that arose in discussion we'd urge the authors to address in the camera ready version. In particular:

1. The authors should clarify whether the approximation in (6) actually converges in any meaningful sense to the posterior expectation in (5). My initial impression is that the answer to this is probably no. While the discussion on lines 195-202 reasonably motivates the use of equation 6, I think motivating this approach through a posterior expectation in equation 5 may slightly oversell the rigor of equation 6, at least as currently described. The authors should consider address whether the approximation is good by running an experiment and comparing their approximation to equation (5) to a monte carlo approximation on a toy scale model where this is feasible, which would more carefully isolate whether (6) is an approximation to (5) or a heuristic.

2. Some of the advantages of the authors' approach are a bit overstated. For example, using SWAG with optimizers other than SGD is fairly common in practice. While obviously this doesn't diminish the authors' results, I think it's worth fixing to ensure technical correctness here in the camera ready.




**Award:**

No

---

### Decision · Program_Chairs · 2022-09-14

Accept